# Individualized discovery of rare cancer drivers in global network context

**Iurii Petrov[1,2], Andrey Alexeyenko[1,2,3]\***

[1]Department of Microbiology, Tumor and Cell Biology, Karolinska Institutet, Stockholm, Sweden; [2]Science for Life Laboratory, Solna, Sweden; [3]Evi-networks, enskild konsultföretag, Huddinge, Sweden

**Abstract** Late advances in genome sequencing expanded the space of known cancer driver genes several-fold. However, most of this surge was based on computational analysis of somatic mutation frequencies and/or their impact on the protein function. On the contrary, experimental research necessarily accounted for functional context of mutations interacting with other genes and conferring cancer phenotypes. Eventually, just such results become 'hard currency' of cancer biology. The new method, NEAdriver employs knowledge accumulated thus far in the form of global inter-action network and functionally annotated pathways in order to recover known and predict novel driver genes. The driver discovery was individualized by accounting for mutations' co-occurrence in each tumour genome – as an alternative to summarizing information over the whole cancer patient cohorts. For each somatic genome change, probabilistic estimates from two lanes of network analysis were combined into joint likelihoods of being a driver. Thus, ability to detect previously unnoticed candidate driver events emerged from combining individual genomic context with network perspective. The procedure was applied to 10 largest cancer cohorts followed by evaluating error rates against previous cancer gene sets. The discovered driver combinations were shown to be informative on cancer outcome. This revealed driver genes with individually sparse mutation patterns that would not be detectable by other computational methods and related to cancer biology domains poorly covered by previous analyses. In particular, recurrent mutations of collagen, laminin, and integrin genes were observed in the adenocarcinoma and glioblastoma cancers. Considering constellation patterns of candidate drivers in individual cancer genomes opens a novel avenue for personalized cancer medicine.

**\*For correspondence:**
andrej.alekseenko@scilifelab.se

**Competing interest:** The authors declare that no competing interests exist.

## Editor's evaluation

In this work, Petrov and Alexeyenko present a novel network-based method to infer cancer driver genes that is not based on frequency of mutations, NEAdriver, and evaluate its performance across a large dataset. This manuscript addresses a topic of high interest in the cancer genomics community and is a welcome addition to the literature.

## Introduction

Carcinogenesis is a complex, multi-step process, during which cellular genomes accumulate new, somatic alterations which might also interact with e.g. germline variants. Mutations that cause or facilitate cancer initiation and progression are called drivers. On the other hand, many mutations occur spuriously due to impairment of chromosome maintenance, replication errors etc. Therefore, a cancer cell genome usually represents a mixture of driver and passenger mutations (*Torkamani et al., 2009*). Apart from the boosting genome instability that generates passengers as well as additional drivers, cancer cells should acquire selective advantages, such as apoptosis evasion, unconstrained

proliferation, or survival in low-oxygen environment which correspond to the hallmarks of cancer (*Hanahan and Weinberg, 2011*). Given the avalanche of new data from cancer genome sequencing, it became possible to complement earlier known, 'core' cancer gene sets with multitudes of computationally inferred drivers.

The existing computational approaches to cancer driver discovery can be classified into three major method groups, possibly combined within a certain implementation:

1. Mutation frequency analyses, based on the idea that driver genes appear mutated more often than expected by chance (*Lawrence et al., 2014*)·(*Mermel et al., 2011*). In order to keep discovering novel drivers, frequency methods should capture increasingly more rare events (*Vogelstein et al., 2013*) – which is limited by practically achievable genomics dataset sizes. As an example, MuSiC driver analysis included only point mutations (PM) that occurred in more than 5% of tumours (*Dees et al., 2012*). A close-to-comprehensive frequency analysis might require 600–5000 samples per tumour type, depending on background mutation frequency (*Lawrence et al., 2015*). Thus, despite all the advancements, cancer sequencing often fails to identify any driver events in a certain cancer genome.
2. Evaluation of functional impact of sequence alterations using protein structural information, physicochemical features, evolutionary conservation etc. (*Reva et al., 2011*)·(*Sim et al., 2012*)·(*Adzhubei et al., 2010*) Such methods might also include frequency analyses and were often trained on smaller sets of best known cancer genes (*Martelotto et al., 2014*) which might lead to overfitting. Although some positive correlation with higher mutation frequency has been demonstrated (*Gnad et al., 2013*), predictions by different methods often disagreed even for most studied genes (*Tamborero et al., 2013*).
3. Commonality of protein function to disease genes established via expert judgement or computational analysis of literature associations (*Jimenez-Sanchez et al., 2001*) and global gene network context (*Torkamani and Schork, 2009; Tranchevent et al., 2011*; *Doncheva et al., 2012*). In contrast to the approaches described above, this 'guilt-by-association' (GBA) methodology (*Oliver, 2000*) did not require information on mutations per se and could thus be applied to all known genes. Most commonly, likelihood of a general function such as cancer 'driverness' was assigned by a GBA algorithm alone which, when applied to all the genes, generated thousands of predictions with prohibitively high false positive rates (FPR).

A particular challenge would be to identify drivers among gene copy number alterations (CNA), which may encompass longer chromosomal regions with multiple genes were gained or lost at once, 'competing' for a driver role assignment. Therefore, CNA genes were often excluded from the analyses described above.

Network analysis is an important tool for cancer driver gene discovery: it not only implements the GBA principle, but also assesses genomic events by employing the network-defined entities, such as modules and pathways. Non-biological algorithms of network analysis, such as PageRank (*Page et al., 1999*) and Random Walk with Restart (RWR), exist since long ago and were adopted, usually with minimal or no changes, by bioinformatics frameworks (*Erten et al., 2011*; *Fang and Gough, 2014*; *Köhler et al., 2008*; *Ozturk et al., 2018*; *Winter et al., 2012*). For a combination of natural and historical reasons, interpretation of these algorithms tend to focus on network hubs, which could miss novel disease genes with lower node degrees (*Barabási et al., 2011*). Furthermore, GBA methods might be work regardless of predefined pathways, for example consider expression correlates (*Torkamani and Schork, 2009*), physical interactions (*Ciriello et al., 2012*), or shared annotations (*Freudenberg and Propping, 2002*). Such methods though, when applied to either all or to frequently mutated genes, would either suffer from the high false positive rate or miss rare drivers. Another solution was offered by the method of network enrichment analysis (NEA)(*Alexeyenko et al., 2012*), where network connectivity is normalized by gene node degrees, which allowed studying genes poorly covered with experimental data. The other advantage of NEA is its high sensitivity and robustness due to considering the multitude of edges available in the global network (*Jeggari and Alexeyenko, 2017*; *Franco et al., 2019*). The concept of enrichment, that is detection of signal that prevails over noise is implemented in NEA via counting network edges that connect gene nodes. Significant excess of actual number of edges over expected by chance can distinguish functionally relevant genes, that is drivers differ from passengers by relevant fragments of network connectivity.

The above-mentioned problem of high FPR can be efficiently addressed by combining probabilities from multiple evidence channels. In the presented analysis we did that in a three-pronged way.

First, we reduced FPR by considering only genes altered in a given tumor genome, whereas genes mutated elsewhere were ignored. Second, we employed the idea that driver mutations of mutated gene sets (MGS) in individual samples should be mutually related and identified such cases by network enrichment against each other, that is within MGS. Third, we detected driver roles by summarizing network connectivity of MGS to diverse potentially informative pathways. Since the full set of such pathways was not known in advance, we started from hundreds pathway profiles, followed by feature selection and creation of predictive cohort-specific sparse models. Combining the two predictors decreased FPR even further.

We applied the analysis pipeline to nine largest TCGA cohorts as well as to a newly compiled meta-cohort of medulloblastoma (MB). We evaluate agreement between our and earlier published driver sets, relative contributions of the driver score components, significance, prediction error rates, and prove the method robustness across over a broad range of mutation rates (from very low in MB to very high in skin melanoma) and variable disease aggressiveness. We demonstrate functional relations between driver mutation patterns, gene expression in affected pathways, and patient survival. Finally, the analysis exposed so far underestimated protein categories with individually rare genomic alterations in their members which appeared essential for several cancer hallmarks.

## Results

### Algorithm outline: two evidence channels for driver prediction

The procedure evaluated likelihood of each genomic alteration reported in MAF or CNA files after level 3 analysis being a driver in the given tumor genome. This was done by considering functional network context in two parallel, independent analysis channels (*Figure 1*):

#### MutSet channel

Evaluated network enrichment between each altered gene $m$ ($m \in$ MGS) and the constellation of all other altered genes $n$ ($n \in$ MGS; $n \neq m$). The resulting NEA scores $Z_{m \leftrightarrow MGS}$ accounted for network degrees of the interacting MGS genes and expressed strength of the cumulative interaction compared to a value expected by chance, that is when $m$ would be functionally unrelated to the rest of MGS.

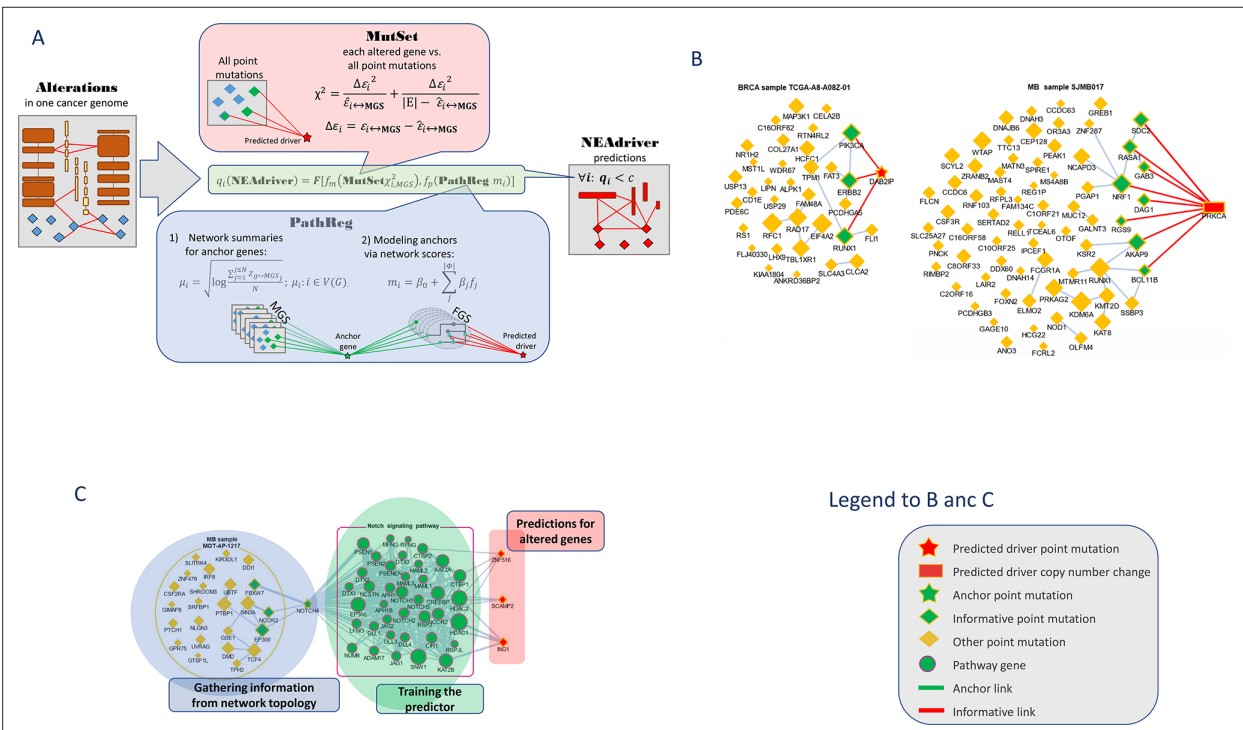

**Figure 1.** Visualization of NEAdriver analysis. (**A**) Workflow according to the algorithm described in Methods. (**B**) Examples of MutSet analysis to mutation gene sets in two cohorts.(**C**) Example PathReg analysis in MB cohort. Legend to nodes and edges.

## PathReg channel

Evaluated likelihood of being a cancer driver in a two-step procedure. First, each of the $N$ genes present in the network ($N$=19035) were characterized – in the same way as in MutSet – by network enrichment scores against each individual MGS. These 'anchor' scores $Z_{i \leftrightarrow MGS}$ were summarized per gene $i$ and within each of the 10 cohorts $c$. The reasoning behind this was that if MGSs included some actual (but not explicitly declared) drivers, then potential cancer involvement of a gene could be expressed as a summary of its network interactions over the MGS collection. For a highly scoring gene, this should become evidence of being a driver when it was altered in a given genomic sample. The cohort-specific gene vectors called 'anchor.summary' should already contain all information gathered from genes' interactions with all the MGSs. However, given that both MGSs and the available network were likely incomplete, some genes in anchor.summary vectors would not be fully evaluated. Therefore, we introduced a second step: boosting via pathway enrichment scores. Cohort-specific predictive models were created using anchor.summary vectors as independent variables (being split into training and testing halves, $N'=N/2$ and $N''=N/2$). Dependent variables for the models were provided by pre-calculated NEA scores for the same $N$ genes against a collection of $P$=320 pathways, generally called functional gene sets (FGS), forming an N×P matrix. Then sparse multivariate models (k=31…83 pathways with non-zero coefficients) were obtained via the lasso training procedure under cross-validation and by controlling performance and robustness with error terms and information criteria. Due to big sample sizes ($N' \sim 10^4$), the models reproduced well on the test sets (Spearman rank R between observed and predicted vectors of anchor.summary were 0.62…0.79; Suppl.File 1). The sparse models produced cohort-specific PathReg scores for each of the $N$ genes. Using published driver sets as references, performance of the scores was compared to the original anchor.summary, which demonstrated clear superiority of PathReg and thus gain in driver-related information via pathway enrichment. Thus, MutSet estimated driverness strictly in the context of individual cancer genomes, whereas PathReg models were originally derived from individual MGSs and then presented as universal, cohort-specific values. Importantly, neither of the two employed information on mutation frequencies. Enrichment heatmaps for genes that scored highest in PathReg versus pathways included in the models are presented in ***Supplementary file 1***.

Similarly to NetSig5000 method (***Horn et al., 2018***), we also tested the approach of network evaluation of mutations against sets of most frequently mutated genes, but this channel did not yield any further advantage and was not used. The MutSet and PathReg scores were calibrated and converted into $p$ and respective $q$ (false discovery rate) values. Evidence from MutSet and PathReg was combined under OR condition, that is the resulting product q(MutSet&PathReg)=q(MutSet)*q(PathReg) reported the probability of NOT being a driver despite positive evidence obtained from either channel. In this way, MGSs were reduced to driver gene sets, DGS (***Figure 1A***). For the purposes of further testing, the DGSs were defined at two significance thresholds q(MutSet&PathReg)<0.05 and q(MutSet&PathReg)<0.01. Under the former cutoff the fractions of drivers were 14…67% larger than

**Table 1.** Fractions of individual alterations and unique genes predicted by NEAdriver.

| | | BRCA | COAD | GBM | LUAD | LUSC | OV | PAAD | PRAD | SKCM | MB |
|---|---|---|---|---|---|---|---|---|---|---|---|
| | genes | 17,216 | 16,553 | 11,484 | 17,028 | 14,853 | 14,055 | 14,766 | 11,660 | 17,253 | 18,244 |
| | samples | 989 | 269 | 284 | 519 | 178 | 461 | 185 | 300 | 346 | 564 |
| No. of | alterations (PM&CNA) | 158,982 | 115,250 | 36,230 | 193,285 | 71,238 | 81,719 | 64,027 | 33,985 | 230,159 | 96,263 |
| | PathReg | 1.55% | 2.23% | 3.81% | 3.02% | 2.94% | 3.81% | 0.16% | 3.44% | 4.42% | 0.06% |
| Fraction of cases when received q<0.05 | MutSet | 2.95% | 3.52% | 3.75% | 4.88% | 3.68% | 3.53% | 1.21% | 2.62% | 9.24% | 4.48% |
| | PathReg & MutSet | 8.52% | 6.81% | 10.63% | 9.67% | 7.72% | 9.15% | 2.79% | 7.74% | 13.42% | 9.14% |
| | PathReg | 0.55% | 1.37% | 1.24% | 2.41% | 2.3% | 2.16% | 0.03% | 2.80% | 3.17% | 0.02% |
| Fraction of cases when received q<0.01 | MutSet | 2.17% | 2.77% | 2.65% | 4.01% | 2.57% | 2.68% | 0.70% | 1.80% | 8.00% | 3.65% |
| | PathReg & MutSet | 7.11% | 5.57% | 7.81% | 8.21% | 6.16% | 7.17% | 1.65% | 6% | 11.81% | 6.98% |
| No. of genes which received q(PathReg&MutSet)<0.05 in>90% samples | | 221 | 343 | 226 | 498 | 334 | 270 | 14 | 180 | 766 | 5 |

under the latter (*Table 1*). The full lists of genes for the ten cohorts are presented in the summary tables (*Supplementary file 8*).

## Comparison with alternative gene sets

We first evaluated performance of the method [q(MutSet&PathReg)<0.05] by ability to detect gene members of 11 alternative reference sets, which were either derived from curated resources or published as computational analysis results (*Figure 2A*). Overlaps with the reference sets were mostly significant: 90 out of 110 pairwise comparisons by Fisher's exact test and 61 out 110 by Mann-Whitney test received a Bonferroni-adjusted p-value below 0.05.

In order to see differences between functional landscapes of NEAdriver (the sets of predicted drivers at (q(MutSet&PathReg)<0.05) vs. the alternative sets), we calculated network enrichment scores of each gene set as a whole versus each of 50 hallmark gene sets (*Liberzon et al., 2015*; *Figure 2B*). The heatmap revealed that NEAdriver detected genes from a different hallmark subspace than most of the computational methods. On the other hand, the NEAdriver predictions often clustered together with the curated sets KEGG05200 'Pathways in cancer' and the union of five 'general' (not cohort-specific) cancer-relevant KEGG gene sets. Similar patterns were observed while looking at the outputs from PathReg and MutSet channels separately, although the former was somewhat closer to the curated sets (Figure Supplements to *Figure 2B*). Compared to the computational methods, the NEAdriver sets and the curated sets showed higher enrichment in EMT, angiogenesis, and suppressed KRAS signaling, glycolysis, inflammatory response, and hypoxia while depleted in cell cycle, DNA replication/repair, peroxisome as well as MYC and mTOR signaling. For comparison, the computational sets were much more similar to the original, full MGSs (Figure Supplement to *Figure 2B*), which confirmed that the NEAdriver pattern was functionally specific and distinct rather than reflected the initial mutation composition. We also noted that nearly all the alternative cohorts (except MutSig) abounded in genes with higher network degree, which were likely better known and studied than the genes predicted with NEAdriver. Node degrees of the latter were closer to an average level, as illustrated by comparisons against random gene samples (*Supplementary file 2*). Remarkably, the network-based method NetSig5000 also prioritized genes with higher node degree.

## Estimation of discovery rates

The same reference gene sets were used for a more detailed evaluation of NEAdriver error rates. The best combination of true positive and true negative rates was found against the gold standard cohort-specific sets, which were either literature-based or available as KEGG pathways (*Figure 3A and B* and upper left plots in Figure Supplements to *Figure 3*), except BRCA cohort, where better results were found for NetSig5000 set which was derived from just this cohort (*Horn et al., 2018*).

Precision was first estimated using the common definition as fraction of true positives among all positives:

$$Precision = \frac{TP}{TP+FP}$$

If applied to the full set of *n* genes, regardless of their mutation status in specific cancer genomes – which corresponded to GBA approach – then these estimates appeared very low and never exceeded 20% at TPR = 10% (upper right plots in Figure Supplements to *Figure 3*).

A more advanced estimate could be provided by using the hybrid positive predictive value (PPV) formula by John Ioannidis (*Ioannidis, 2005*), where the frequentist terms – error rates of types I and II – were combined with odds (i.e. the ratio of actual drivers versus non-drivers among the mutated genes), which represented a Bayesian component:

$$PPV = \frac{(1-\beta)*R}{(1-\beta)*R+\alpha}$$

The odds could be estimated from the union of all test results on the cohort's MGSs as

$$R = \frac{TP+FN}{FP+TN}$$

By expanding the Bayesian approach, the type I and type II errors would be, respectively: $\alpha = \frac{FP}{FP+TN}$ and $\beta = \frac{FN}{TP+FN}$.

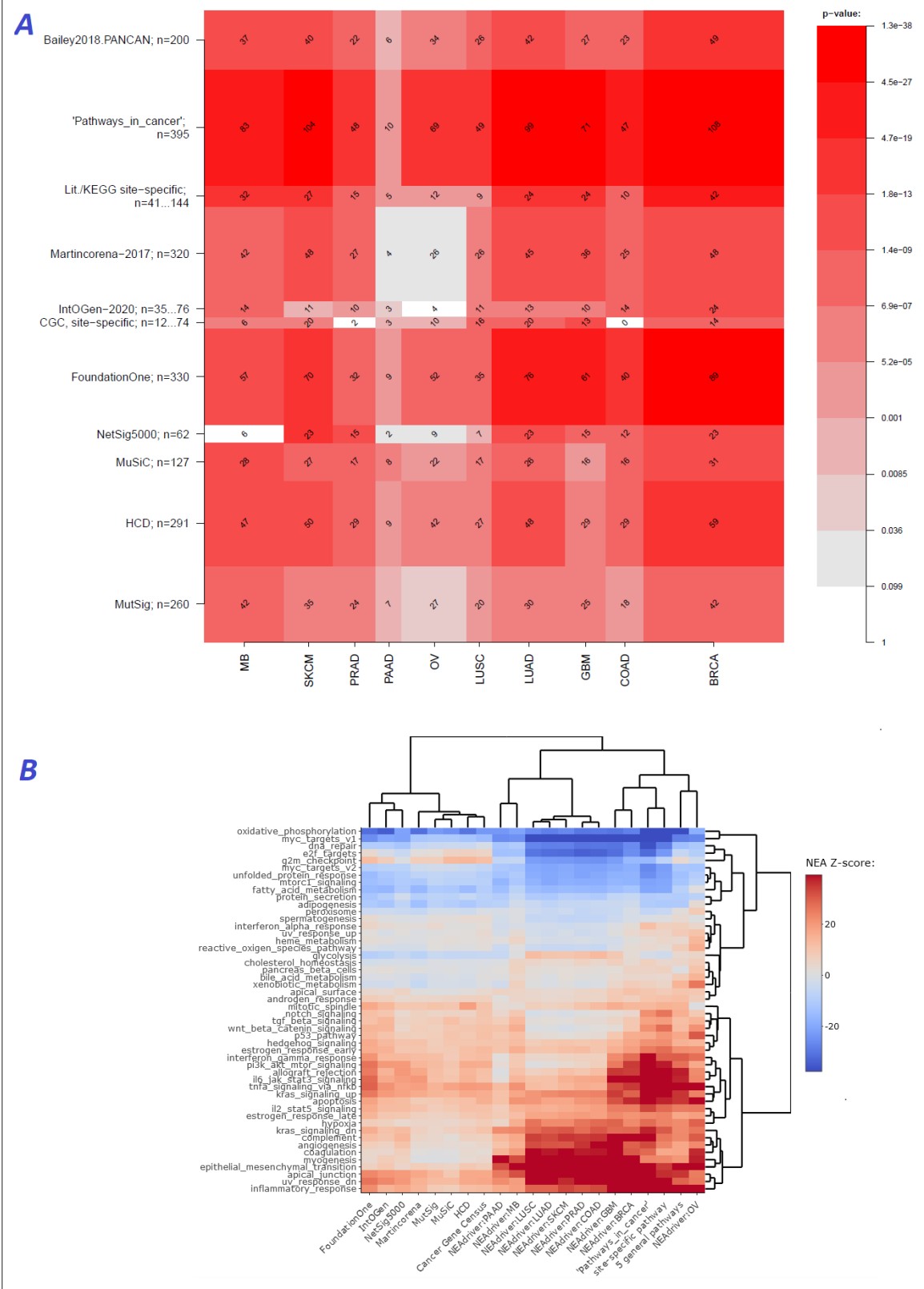

**Figure 2.** Agreement between NEAdriver and reference gene sets. (**A**) The heatmap matrix elements represent overlap between the cohort-specific sets of predicted NEAdriver gene sets at q(MutSet&PathReg)<0.05 and gene sets from curated resources and alternative methods. Row and column widths are proportional to gene sets sizes. All the reference gene sets had fixed, 'pan-cancer' member sets independent of cancer site, except Cancer Gene Census, IntOGen, and literature/KEGG site-specific sets, for which size ranges are given. B. Network enrichment of the cancer gene sets with

*Figure 2 continued on next page*

*Figure 2 continued*

regard to 50 hallmarks (*Liberzon et al., 2015*). NEAdriver sets defined at q(MutSet&PathReg)<0.05 are represented by 165 genes for each cohort, most frequent across its samples(n=165 was chosen for being half of the size of FoundationOne set).

The online version of this article includes the following figure supplement(s) for figure 2:

**Figure supplement 1.** Network enrichment of the cancer gene sets with regard to 50 hallmarks.

**Figure supplement 2.** NEAdriver sets defined at q(MutSet)<0.05 and represented by 165 genes for each cohort.

**Figure supplement 3.** NEAdriver sets defined at q(PathReg)<0.05 and represented by 165 genes for each cohort.

**Figure supplement 4.** Correlations between from PathReg and MutSet channels.

The value of FP + TN could be approximated as the total number of genes, since the number of drivers (TP and FN) should be negligible compared to that.

However, estimating the TP and FN via reference (gold standard) sets was more challenging, since the source publications and databases never claimed that their gene sets are truly complete. Thus, PPV estimates were particularly sensitive to biases in TP and FN and we therefore tried each of the nine sets. PPV ranged from 30% to 70% at TPR = 10%, but even at TPR = 100% almost never dropped below 20% (*Figure 3C and D* and all cohorts in Figure Supplements to *Figure 3*). Again, the best performance was achieved using the literature/KEGG sets (PPV = 44…68% at TPR = 10%).

Since this approach considered any genes not listed in each given set as false findings, the PPV estimates must have been excessively conservative. Therefore, we next investigated the potential of discovering novel drivers using genes collected from site-specific literature or respective KEGG pathways. An alternative, pan-cancer estimate was made with a set of 369 'known cancer genes' (*Martincorena et al., 2017*). Applying the PPV adjustment to these sets under assumption that they were just 50% complete increased the PPV estimate by 20…30% (dotted curves at *Figure 3E and F*). Recalling that (1 − PPV) is essentially synonymous to false discovery rate (i.e. q-value) allowed us also to compare error rate estimates from the two independent approaches: the gold-standard based PPV versus the continuous NEAdriver q(MutSet&PathReg). Although the relation was not linear over the range PPV = 0…100%, at PPV = 30…70% the cancer site-specific estimates were remarkably close in each of the 10 cohorts. On the other hand, the pan-cancer benchmark of 369 known (mostly computationally) cancer genes estimated NEAdriver q as inflated by 20…40% while assuming that the gene set was complete (solid lines) but well matching the PPV while assuming 50% incompleteness (dotted lines) (*Figure 3E and F* and Figure Supplements to *Figure 3*). Therefore, we used NEAdriver q(MutSet&PathReg) for reporting confidence of driver predictions in this work.

We also compared the results to a number of previously suggested network-based methods that considered impact of somatic alterations on the transcriptome: DriverNet (*Bashashati et al., 2012*) and HotNet2 (*Leiserson et al., 2015*) which implemented the cohort level approach as well as SCS (*Guo et al., 2018*), OncoIMPACT (*Bertrand et al., 2015*), and DawnRank (*Hou and Ma, 2014*) which worked at the individualized, single-patient level. The comparison also included naïve frequency-based estimates as provided by Guo and co-authors (*Supplementary file 3*). These publications presented short candidate driver lists combined over all samples. Agreement of the integrated ranks with NEAdriver confidence q(MutSet&PathReg) in the four TCGA cohorts proved to be significant albeit rather weak (Spearman $R$=0.22…0.30). Further, we focused on lists of top 50 genes from each of the methods. In three cohorts (OV was the exception), a good agreement was found between the methods and NEAdriver (*Supplementary file 4*). Out of top 50 driver lists, between 9 and 41 genes received NEAdriver q(MutSet&PathReg)<0.05 (the overlaps were significant after Bonferroni-adjusted Fisher's exact test $p<0.001$).

## Rates of driver discovery versus mutation frequency and possible confounders

Driver discovery from large-scale genome sequencing might produce false positives due to various biasing factors, such as gene length, transcriptional activity, or DNA replication rate (*Lawrence et al., 2013*). Although NEAdriver was designed to be independent from alteration frequency – in order to be thus more sensitive to rare events – we still performed in-depth analysis of NEAdriver output in relation to such factors and in comparison with alternative methods.

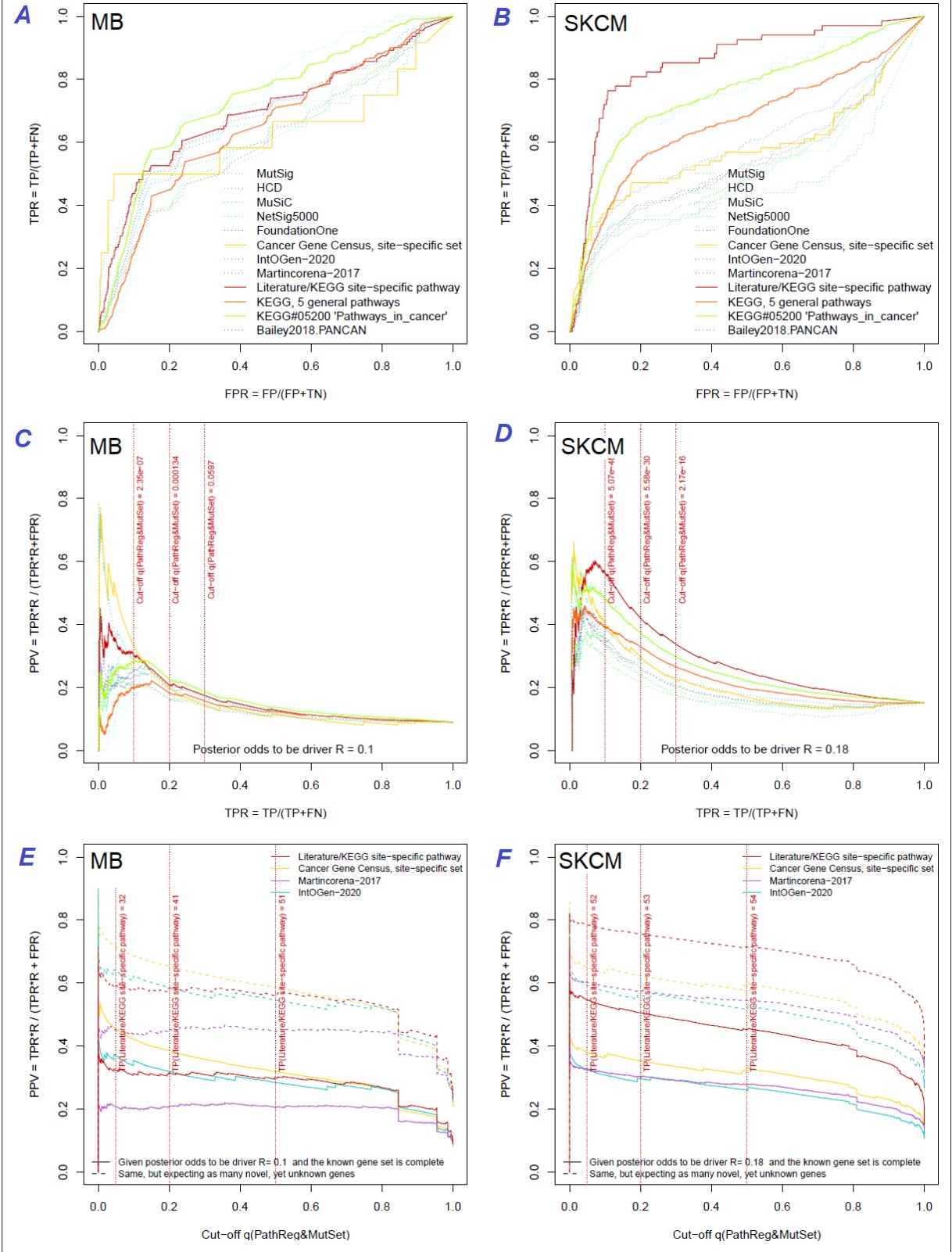

**Figure 3.** Performance of the new driver prediction evaluated on different benchmarks. Two cohorts with very low versus high passenger mutation load, medulloblastoma (MB: **A,C,E**) and skin cutaneous melanoma (SKCM: **B,D,F**), represent contrast conditions for computational driver discovery. The NEAdriver predictions were quantified by the cumulative statistic q(MutSet&PathReg)<0.05 and matched to reference sets. (**A**) and (**B**): ROC curves in the space of true positive versus false positive rates in the classical definition of 'precision'. (**C**) and (**D**): Precision-recall curves where precision was

*Figure 3 continued on next page*

*Figure 3 continued*

calculated via inclusion of odds 'driver/non-driver'.(**E** and **F**) calibration of positive predictive value, PPV against false discovery rate (q~1 – PPV; solid lines) and modeling of PPV in presence of true, but yet unknown drivers (dot-dashed lines) using site-specific and pan-cancer benchmarks. The dotted vertical cutoff lines referto cancer site specific pathway sets, taken from either to the literature or respective KEGG pathway. Cutoffs in (**C**) and (**D**) display q(MutSet&PathReg) values, whereas TP counts in (**E**) and (**F**) are numbers of unique site-specific genes discovered under variable q(MutSet&PathReg) threshold shown at X-axis.

The online version of this article includes the following figure supplement(s) for figure 3:

**Figure supplement 1.** Performance of the new driver prediction evaluated on different benchmarks.

**Figure supplement 2.** Performance of the new driver prediction evaluated on different benchmarks, all cohorts.

**Figure supplement 3.** Performance of the new driver prediction evaluated on different benchmarks, PRAD.

**Figure supplement 4.** Performance of the new driver prediction evaluated on different benchmarks, PAAD.

**Figure supplement 5.** Performance of the new driver prediction evaluated on different benchmarks, OV.

**Figure supplement 6.** Performance of the new driver prediction evaluated on different benchmarks, LUSC.

**Figure supplement 7.** Performance of the new driver prediction evaluated on different benchmarks, LUAD.

**Figure supplement 8.** Performance of the new driver prediction evaluated on different benchmarks, GBM.

**Figure supplement 9.** Performance of the new driver prediction evaluated on different benchmarks, COAD.

**Figure supplement 10.** Performance of the new driver prediction evaluated on different benchmarks, BRCA.

The genes listed by FoundationOne, MutSig, HCD, MuSiC and in particular by Cancer Gene Census and IntOGen had generally more mutation events per cohort than drivers predicted at q(MutSet&PathReg)<0.05 (*Figure 4*, left panes). The same tendency was observed when comparing copy number altered genes predicted by NEAdriver (*Figure 4—figure supplements 1–10* to *Figure 4*). Exceptions could only be found in SKCM cohort (which was not analysed in most of these projects) and in a few cohorts for MutSig (which implemented advanced normalization approaches). Sensitivity of NetSig5000 to rare mutations was comparable to our method – likely due to its network-based approach – but again mostly to genes with higher network degree (see details in *Figure 4—figure supplements 1–20* to *Figure 4*). On the contrary, when mutation frequencies were normalized by coding sequence length, the differences between the methods became less pronounced (*Figure 4*, right panes). This was not surprising, since shorter genes manifest mutations less frequently.

Lawrence and co-authors (*Lawrence et al., 2013*) demonstrated that simply considering mutation frequency per gene without accounting for genomic factors results in multiple false positive associations. Similarly to their Figure 3 , we evaluated influence of suggested confounders, namely gene length, replication rate, expression level, and total mutation burden per sample on the NEAdriver predictions. Each of the linear models included one of these confounders together with number of mutations per cohort per gene as well as being a known driver or a known artifact. While the mentioned *Figure 3* by Lawrence and co-authors showed strong correlations of mutation rate versus replication or expression, respective correlations of NEAdriver score were not strong at all (albeit formally significant; absolute values of Kendall tau <0.05; *Supplementary file 5*). NEAdriver predictions were stronger associated with gene length (absolute values of Kendall tau = 0.07…0.15) – while we also noticed such association in all the alternative gene sets (*Figure 4—figure supplement 21* to *Figure 4*). Also, genes of the latter sets contained point mutations more frequently and had higher mutation frequency per b.p. length, both absolute and normalized by genome-specific mutation load. This also characterized the driver set by *Lawrence et al., 2014*, which would supposedly be least affected by these factors due to inclusion of these covariates in their models. Otherwise, probability of being predicted by NEAdriver after adjustment for gene length and TMB proved to be weak (Kendall tau <0.05 in seven out of ten cohorts; *Supplementary file 5*). We also note that the replication and expression analyses must vary between tissues and datasets, while being expensive to measure (e.g. the analyses by Lawrence and colleagues used data from less than 100 cell lines).

Finally, we specifically considered 'artefactual' or 'false positives' genes that had cropped up in earlier cancer genome studies and were explicitly listed in later literature (34)(40)(41). We also included olfactory receptor genes as a whole category, supposedly prone to artifacts (although a majority these lacked any network edges and could not produce non-zero NEA scores). Among NEAdriver predictions, the fractions of any artefactual genes were much smaller than the overall false discovery rate

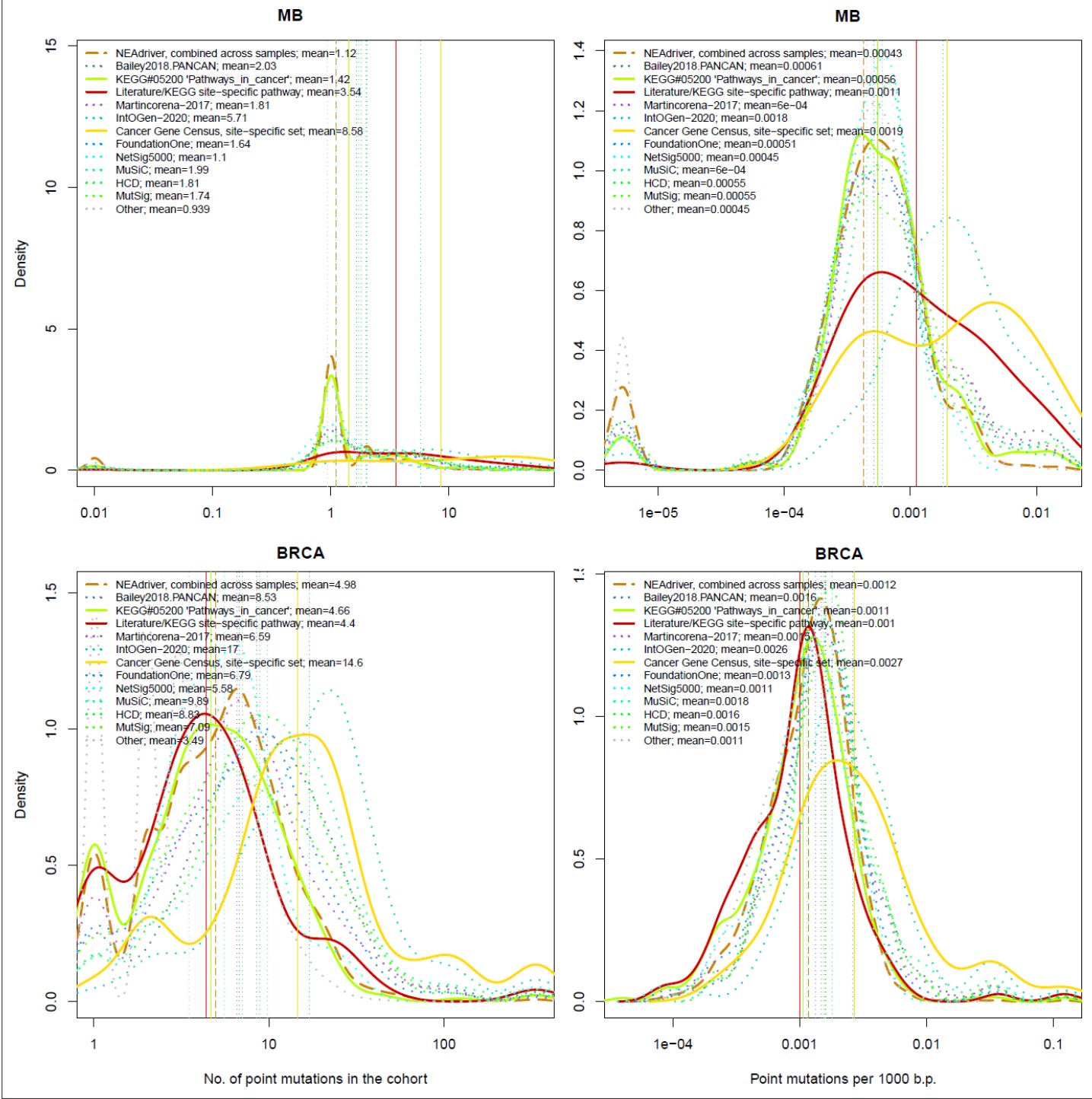

**Figure 4.** Comparative analysis of point mutation frequency among genes included in cancer gene sets. Density plots shape the distributions in each of the alternative sets, predictions by NEAdriver (q(MutSet&PathReg)<0.05; brown dashed line), and genes not included in any of the above ('other'; gray dotted line). Vertical lines correspond to mean values provided in the legend.

The online version of this article includes the following figure supplement(s) for figure 4:

**Figure supplement 1.** Comparative analysis of point mutation frequency among genes included in cancer gene sets.

**Figure supplement 2.** Performance of the new driver prediction evaluated on different benchmarks, SKCM.

**Figure supplement 3.** Comparative analysis of point mutation frequency among genes included in cancer gene sets, PRAD.

**Figure supplement 4.** Comparative analysis of point mutation frequency among genes included in cancer gene sets, PAAD.

*Figure 4 continued on next page*

DOI: https://doi.org/10.7554/eLife.74010

*Figure 4 continued*

**Figure supplement 11.** Ten cohort files with density plots shaping mutation frequency distributions for genes in each of the alternative sets, predictions by NEAdriver (brown dashed line), and genes not included in any of the above ('other'; gray dotted line).

**Figure supplement 21.** Boxplots comparing lengths of genes included in different sets versus rest of known genes.

evaluated via either q-value or PPV as described above. Only between 0.5% and 3% of the predictions were found in the artifact gene lists (upper right legends in new *Supplementary file 5*). In the cohort-specific linear models (bottom left legends), the Bonferroni-adjusted p-values for the term 'known artifact' were lower than 0.05 only in five cases out of the 30. The artefactual genes are text labelled in the scatterplots when surpassed significance threshold of q=0.05 (1…13 genes per cohort) and marked in the last columns of the summary tables (*Supplementary file 8*). For comparison, these genes made up 1…9% of any other computational set in our analysis.

## Novel findings

How many known drivers there are in individual cancer genomes and by how much the new method could expand this space? An earlier computational analysis estimated the number of point driver mutations as two to six per genome (*Kandoth et al., 2013*). In our study – by counting any genes included in the nine alternative sets (N=1434) – the modes (most frequent count values) ranged across the cohorts between $M$=1…3 in MB (known to have very low somatic mutation load) to M=55 in SKCM (having typically thousands mutated genes per sample). For NEAdriver [q(MutSet&PathReg)<0.05], respective values per genome were lower, ranging between $M$=0…1 (MB) to $M$=50 (SKCM) (*Figure 5A*). Overlaps between these two approaches were rather modest ($M$=0…8). In other words, the driver candidates identified by NEAdriver were mostly novel. The overlaps between 'alternative sets' and NEAdriver [q(MutSet&PathReg)<0.01] are also presented for individual cancer genomes (*Figure 5B*). The Jaccard coefficient values, with exceptions of MB and GBM, rarely exceeded 0.3, which confirmed that NEAdriver identified mostly novel genes.

Recalling that respective PPV estimates reached 50% and exceeded 75% when allowing for novel drivers, the predictions appeared fairly confident.

## Clustering patient driver sets in pathway space revealed association with survival

One of the goals of tumour molecular profiling is to discover cancer subtypes which would be informative of disease outcome or clinically meaningful otherwise. The driver genes identified with our method were mostly rare and therefore not suitable as stand-alone subtype markers. However, using NEA we could generate 'DGS vs. FGS' scores which summarized signals from various disparate events and thus available for every patient. We explored if DGS profiles in the FGS space could partition the cohorts by differential survival.

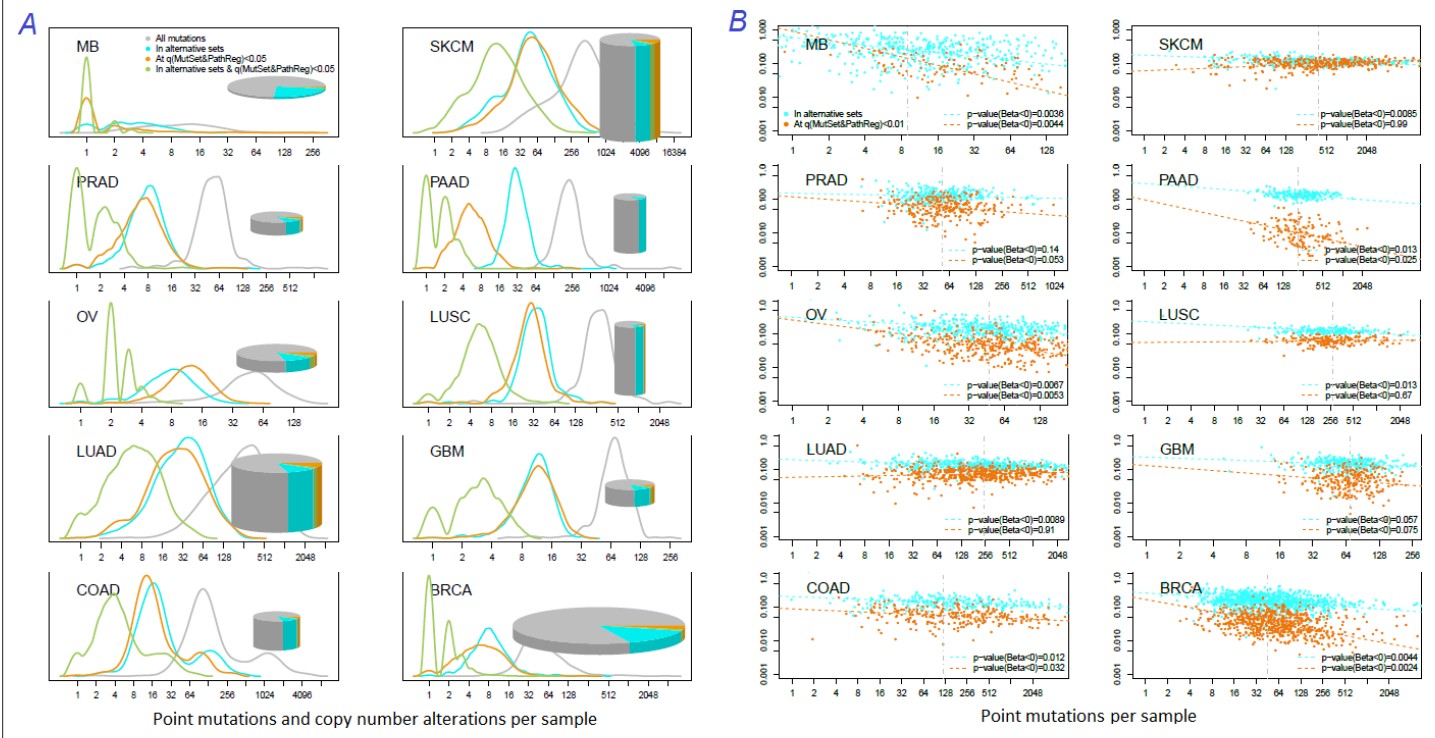

**Figure 5.** Distribution of somatic mutations versus drivers across genomic samples. (**A**) Relative density plots of mutations and declared drivers. Pie charts summarize counts per genomic sample in each of the ten cohorts (height: average number of reported mutations per sample; width: number of samples in the cohort). (**B**) Overlap between the predictions by MutSet&PathReg and the merge of alternative gene sets (1434 genes in total) color by Jaccard index (sets' intersection divided with sets' union). The MGS sizes (regardless of driver status) are expressed as marker size. Gaussian noise was added to marker coordinates for better readability.

Indeed, the DGSs [q(MutSet&PathReg)<0.05] were often informative on patient survival. We tested three different clustering techniques and found that in many cases DGS scores differentiated cohorts by survival: 7…14.8% of all tests yielded significant Cox proportional hazard models (Benjamini-Hochberg FDR <0.25). Furthermore, in up to 21.1% of all tested cases the significant partitions were recapitulated on test sets (while FDR estimates from Cox models were below 0.25) (see examples in *Figure 6* and full details in *Supplementary file 6*). For comparison, splitting in the same framework by high vs. low tumor stage did not differentiate patients by survival (not shown).

## NEA scores based on either drivers or gene expression point to same pathways associated with survival

Finally, we checked if association of specific FGS scores with survival could be traced at the level of mRNA transcription. To this end, we derived lists of 100 patient-specific genes with expression most deviating from the cohort mean (gene expression based AGS) and looked if their NEA scores for the same FGS would also be associated with survival. By testing the 10 cohorts, 2 survival types, 3 clustering methods, and the 1659 FGSs, we identified 31 cases where the association with survival was observed for both DGS-FGS and gene expression based AGS-FGS scores (*Figure 6—figure supplement 1* to *Figure 6*). The discovery of this many associations was significant in a random permutation test requiring Bonferroni-adjusted p-value <0.01 while permutation test-based p-value <0.0001 (*Figure 6—figure supplement 2* to *Figure 6*). Remarkably, in most of the cases opposite relations with survival between DGS and gene expression based AGSs were observed: better outcome was associated with high scores of the former while lower scores of the latter, or vice versa.

For example, MB and LUAD cohorts were differentiated by survival using NEA score profiles for two pathways (*Figure 7*). The cohort patients were represented first by DGSs (left) and then by gene expression based AGSs (right). The NEA scores reflected connectivity between pathway genes and patient genes (either DGS or gene expression based AGS). A higher NEA score would indicate that

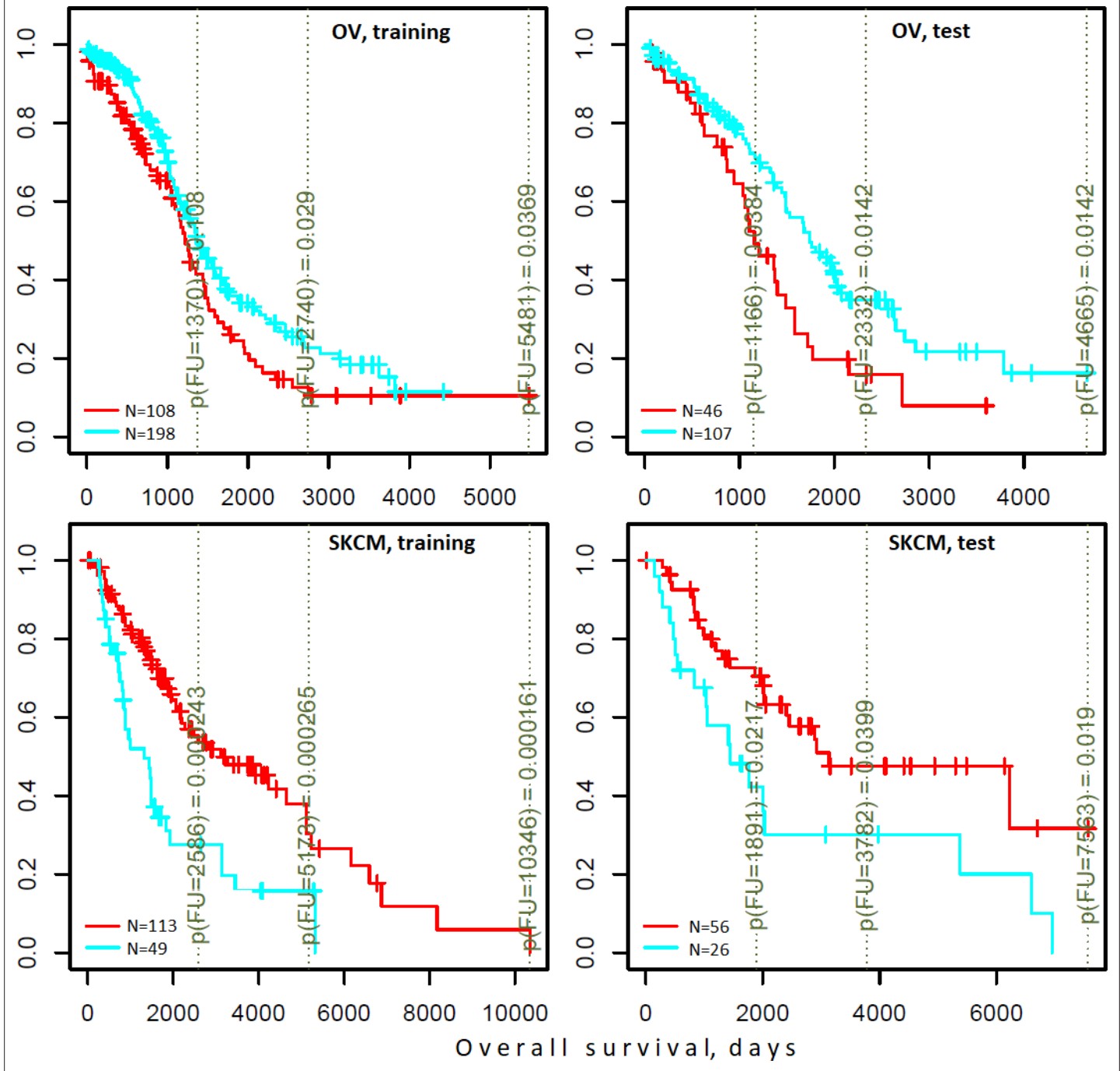

**Figure 6.** Differential survival of patients stratified in pathway space created by network enrichment analysis of driver gene sets. Vertical captions (brown) convey Cox proportional hazard p-values for three follow-up intervals.

The online version of this article includes the following figure supplement(s) for figure 6:

**Figure supplement 1.** Analysis of significance across survival curves.

**Figure supplement 2.** Distribution of p-values for survival correlations from different clustering methods and agreement of P-values on train versus test data sets.

relatively many patient-specific genes were linked to the given pathway. MB cells are known to some-times produce granulocyte colony-stimulating factor (**Pietsch et al., 2008**), which can affect influx of granulocytes (**Vermeulen et al., 2017**) and disease prognosis (**Paul et al., 2020**). With regard to 'Biocarta granulocytes pathway', the MB patients where stratified so that higher DGS scores indicated

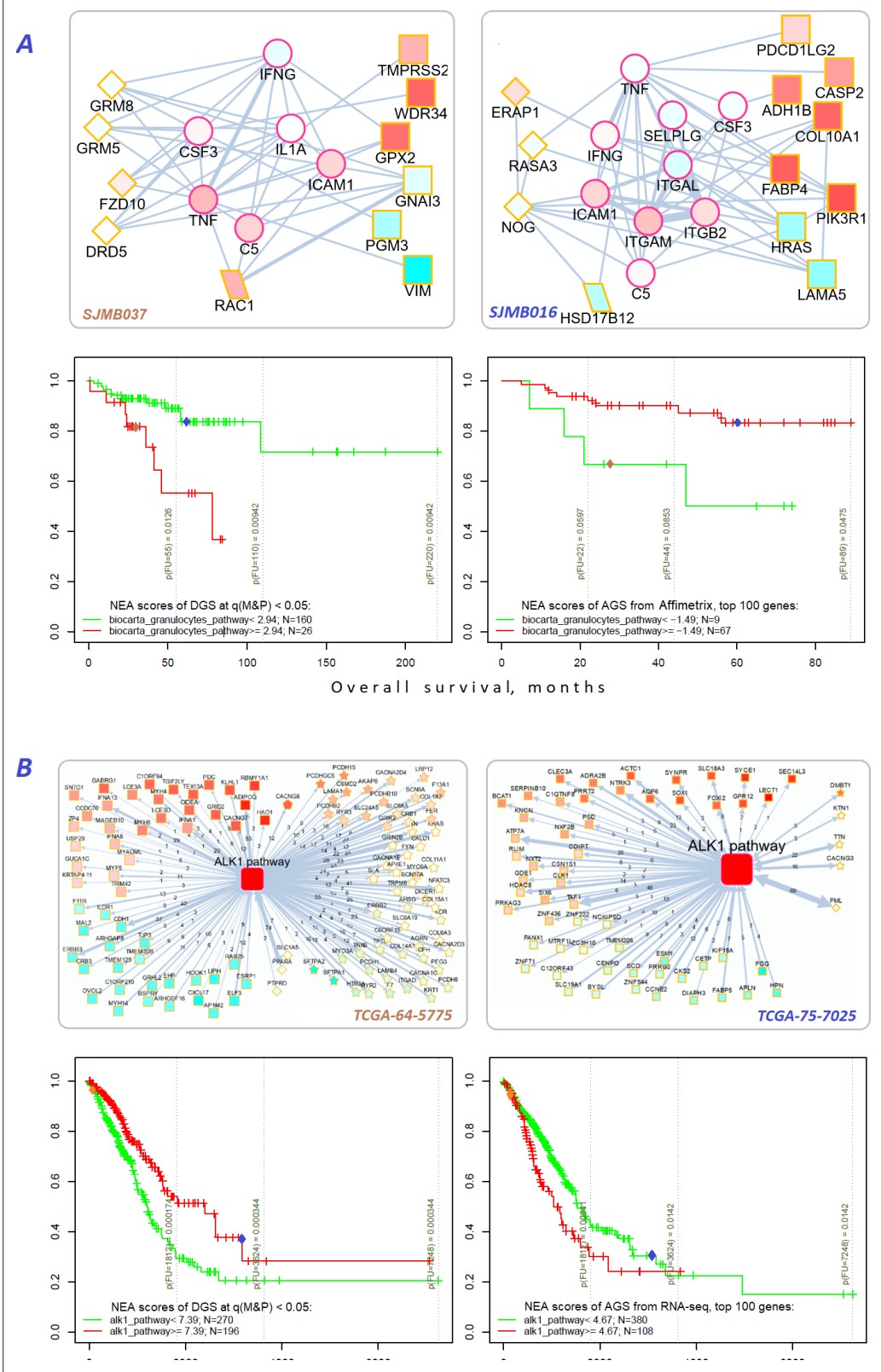

**Figure 7.** Network enrichment and survival analyses of patient specific lists of drivers and differentially expressed genes. (**A**) Example from MB cohort. (**B**) Example from LUAD cohort. Yellow borders: patient-specific gene sets including (*Torkamani et al., 2009*) driver alterations (q(*MutSet&PathReg*)<0.05): either point mutations (stars) or copy number changes (diamonds) (*Hanahan and Weinberg, 2011*) genes with mRNA expression most deviating

*Figure 7 continued on next page*

*Figure 7 continued*

compared to the rest of the cohort (rectangles) (*Lawrence et al., 2014*) both categories 1 and 2 (rhomboids). Magenta borders: pathway genes (circles). Each gene is colored by expression in the given patient sample compared to the cohort mean. Note that pathway genes usually did not manifest genomic or strong expression changes. In figure (**B**) the edges combine individual network links between genes. Links within pathway not shown. Clinical and NEA data for the patients.

The online version of this article includes the following source data for figure 7:

**Source data 1.** Clinical survival data and NEA scores for the MB and LUAD patients.

poorer survival, whereas higher gene expression based AGS scores were associated with better survival. Subnetwork patterns for two patients exemplify this analysis (*Figure 7A*). ALK fusion events are a well-established target for non-small cell lung cancer therapy (*Ross et al., 2017*). While none of the patients were treated with an ALK inhibitor in LUAD cohort, 'Biocarta ALK1 pathway' scores for both DGS and gene-expression-based AGS were informative on overall survival within 6-year follow-up interval. Again, relations to survival were opposite for DGS versus gene-expression-based AGS scores.

## Novel categories of cancer driver genes

We noticed that a large part of the connectivity with regard to functional hallmarks (*Figure 2B*), which distinguished the NEAdriver predictions from other computational gene sets, was due to multiple collagens, laminins, and integrins predicted in most of the cohorts. These genes are typically rather long, within the 2nd quartile of protein coding gene list ranked by CDS length. Their median mutation frequencies per base pair of CDS length were just 1.5…2.5 times higher than that of all protein coding genes, which likely explains their escape from computational analyses so far. Nonetheless, across the ten studied cohorts genomic alterations occurred on average in 2–7 genes of these families per sample (*Figure 8A*). Using logistic regression with total mutation burden per sample as a covariate, we found that point mutations patterns of these genes also significantly (at FDR <0.05 after adjustment for multiple testing) co-occurred pairwise: there were e.g. 84, 10, 23, and 308 such pairs in BRCA, COAD, LUAD, and PAAD cohorts, respectively.

*Figure 8B* displays a typical subnetwork of genes that encode collagens, laminins, and integrins interconnected with heparan sulphate, fibronectin as well as a few signaling proteins – all affected with point mutations or copy number changes in the same cancer genome. This pattern explains high enrichment in network links to epithelial mesenchymal transition, apical junction, and angiogenesis hallmarks presented in *Figure 2B*. Previously, roles of these families in for example cell migration, epithelial mesenchymal transition, or angiogenesis were rather well characterized at the structural (*Ahmed et al., 2005*; *Rousselle and Scoazec, 2020*; *Moilanen et al., 2017*) and tissue-specific transcriptional (*Bretaud et al., 2020*; *Mammoto et al., 2013*) levels, and even suggested as markers for cancer diagnostics (*Risteli et al., 2014*) and targets for treatment (*Tsuruta et al., 2008*). However, they were not recognized by computational analyses, with a few exceptions: COL1A1 (34), COL5A1, COL5A3, ITGB7 (13), COL18A1 and ITGA6 (42), and none were so far included in the Cancer Gene Census or FoundationOne targeted sequencing panel.

## Discussion

So far, most of projects presenting novel cancer drivers s generalized the discovery: either globally, within the pan-cancer paradigm (*Campbell et al., 2020*; *The Cancer Genome Atlas Research Network et al., 2013*) or within site/organ specific tumour types (*Berger et al., 2018*), sometimes delineating subtype-specific drivers (*Pugh et al., 2012*; *Sweet-Cordero and Biegel, 2019*). Such approaches possess lower statistical power regarding short genes. We found that shorter genes were underrepresented in all alternative sets considered in this study, except the curated cancer pathways (Figure Supplement to *Figure 4*). Meanwhile, even rarely mutated genes can be drivers for example in absence of alterations in a 'major' gene, such as TP53 – but identifying such associations would require genome-wide studies at an unaffordable scale (*Stracquadanio et al., 2016*). This situation apparently contradicts the individualized approach to cancer treatment, which suggests molecular pathological analyses for disease prognostication, administration of targeted drugs (*Remke et al.,*

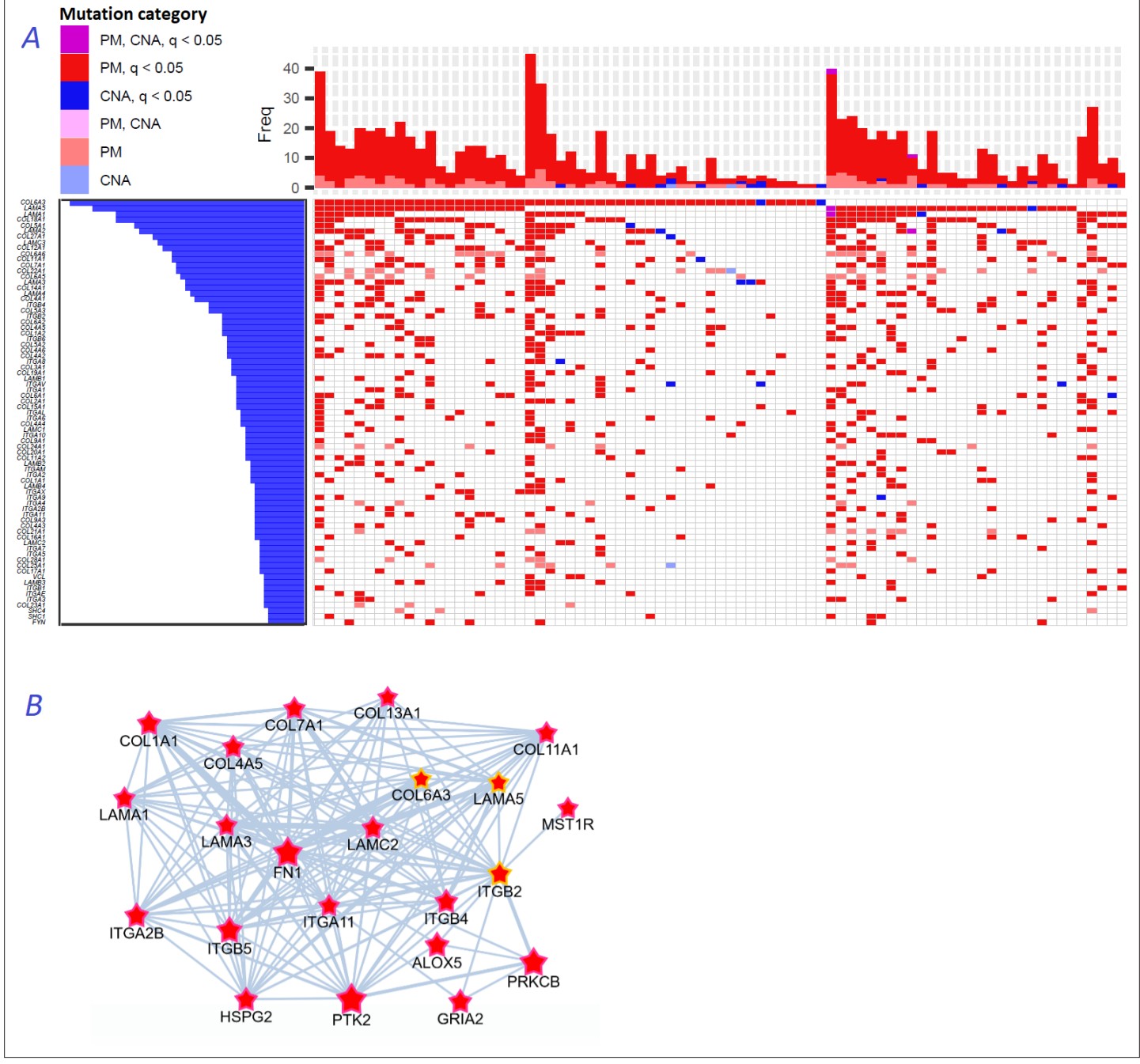

**Figure 8.** Novel gene families in cancer driver context. (**A**) Upper left fragment of a waterfall plot containing top 75 genes for collagens, laminins, integrins, and a few signaling proteins most frequently mutated in COAD cohort (269 samples in total). Genes with point mutations (PM) and copy number alterations (CNA) are colored according to gaining significance as q(MutSet&PathReg)<0.05 or not. (**B**) Point mutations connected with each other in the global network and identified as potential drivers in one genomic sample TCGA-CM-6171–01 (COAD) at q(MutSet&PathReg)<0.05. In particular, point mutations in COL6A3, ITGB2, and LAMA5 were also detected as significantly co-occurring across COAD cohort in a general linear model accounting for total mutation burden as covariate (FDR <0.01).

*2013*), and discovery of novel drug targets. Currently, the majority of patients are not amenable to any approved targeted treatment since respective matching mutations occur with low prevalence. Further development of precision cancer medicine requires considering functional context of cancer genome in each patient (*Wheeler and Wang, 2013*).

Novel approaches to driver identification is therefore urgent: while every cancer genome is expected to possess driver mutations, many cases lack any alterations in known cancer genes – which

is counterintuitive and undermines the ground for targeted therapy. Due to the high mutational heterogeneity of cancer samples, frequency-based methods have reached their limits of statistical power to detect novel cancer drivers.

One should distinguish between mutation frequency as a tool to discover driver genes and the biological mechanism via which driverness is implemented. A mechanism for a given gene would be implemented at the level of individual tumors and therefore does not have to be directly associated with cohort-level statistics. Unlike of most of the methods (including many network-based ones), NEAdriver itself did not lean on the mutation frequency features. Instead, individual mutation events became input to NEAdriver and were evaluated independently of each other.

We pursued this approach via network analysis of mutated genes, by which patient-specific driver constellations should be discerned from the background of passenger burden. The network analysis – an already popular method to identify cancer genes using functional context – in our implementation was relatively less biased toward network hubs and thus more sensitive to novel driver genes. The previous guilt-by-association analyses (*Köhler et al., 2008*; *Cava et al., 2018*; *Cho et al., 2016*; *Reyna et al., 2020*) predicted gene function based on functional connections to known functional categories. Confidence of such predictions alone, for example in absence of experimental data was very low due to rare occurrence of actual mutations and thus lower true discovery rate, as shown by John Ioannidis (*Ioannidis, 2005*). NEAdriver was more focused due to considering concrete molecular phenotypes (*de facto* alterations in individual genomes) and combining relevant evidence from two network analysis channels. MutSet channel possesses an extra advantage: it can be directly, without pre-training on whole-cohort data, applied to a sequencing dataset of a single novel patient.

Since capabilities of both MutSet and PathReg could only be implemented on driver constellations of sufficient size, it was important to test NEAdriver on cancer genomes with different mutation loads. This feature varied from a few affected genes in MB to thousands in lung and skin cancers. Despite the variability of this and other biological parameters, both statistical performance and candidate drivers' functional profiles proved to be rather close across the 10 cohorts. As an example, three pathways were systematically included in the models: hsa04020:Calcium_signaling_pathway (CS), hsa05412:Arrhythmogenic_right_ventricular_cardiomyopathy_(ARVC), and hsa05414:Dilated_cardiomyopathy (DCM). Although the role of CS in cancer was rarely considered central, cell migration and adhesion do involve modulation of cell motility and shape where ion channels and pumps play major roles, so that CS genes are known for both downregulation and functional implication in cancers (*Tajada and Villalobos, 2020*; *Phan et al., 2017*; *Litan and Langhans, 2015*). ARVC and DCM are functionally close to CS, although have little overlap in member genes. At the first glance, the major factor behind MGS-CS interrelations might be the frequently mutating titin TTN. Its involvement in cardiopathies and cancer has been long argued because of its extremely long coding sequence, thus likely prone to spurious alterations (~50% of LUAD samples). However, there were many individually rare mutations, which together revealed emergent network patterns between MGSs and either CS, or ARVC, or DCM: cadherins, laminins, integrins, metalloproteases, nitric oxide synthases, ryanodine receptors, adenylate cyclases, subunits of protein kinase A etc. They contributed to the NEA scores with multiple network links so that for example the median edge counts between genes of MGSs and of the pathways were 22…45 in MB and 497…890 in LUAD. We therefore did not exclude genes commonly supposed to be 'artefactual' but instead labelled them in the final tables.

The cohort- and sample-specific NEAdriver predictions agreed well with nearly all tested alternative sets (*Figure 2A*). The best agreement was found for cancer-site-specific gold standard sets while less so for pan-cancer sets. The computational, mostly frequency-based sets performed worse than curated sets (*Figure 3A and B*). In the functional space, NEAdriver predictions were positioned differently compared to computational and database sets, but close to curated cancer pathway sets (*Figure 2B*), which confirmed both novelty and relevance of NEAdriver findings. The latter also differed from most of the alternative sets in having much less bias in regard of network node degrees and gene length.

A realistic estimate of the true discovery rate for NEAdriver predictions was obtained by accounting for putative drivers not included of the gold standard sets, so that PPV could be as high as 78%...88% at the q(mutSet&PathReg)=0.05, which was verified by the two alternative ways of PPV calculation using four different gold standards (*Figure 3E and F*). The efficiency of NEAdriver was confirmed by

the ability of DGS to stratify patients by survival (*Figure 6*) and the striking tendency of same pathways being associated with survival via both mutation- and expression-based patient scores (*Figure 7*).

Obviously, NEAdriver alone would miss events detectable by other methods, for example when certain 'stand-alone drivers' impose strong effects on their own (TP53, APC etc.), without apparent interaction with other genes. This creates an incentive for creating a combined methodology and a toolbox in the future. But already now our analysis identified hundreds of somatic gene alterations that had not been deemed functional in previous research. After evaluation performed in multiple ways, the body of predictions appears confident, providing a set of provable research hypotheses and suggesting new strategies for cancer prognosis and individualized treatment.

# Materials and methods
## Medulloblastoma meta-cohort
We collected data from publications presenting large-scale datasets (*Jones et al., 2012*; *Northcott et al., 2017*; *Pugh et al., 2012*; *Robinson et al., 2012*) and two public datasets available online (PBCA-DE and PEME-CA). We retrieved available exome sequencing profiles as well as copy number alterations, gene expression, and clinical data. We translated gene identifiers into gene symbols according to ENSEMBL annotations v.93 and then made sure all the gene symbols are found in the network and are up to date according to GeneCards (*Stelzer et al., 2011*) annotations.

For consistency with the publication datasets, we excluded the following types of mutations from PBCA-DE and PEME-CA sets: intron variant, upstream gene variant, 3_prime_UTR_variant, 5_prime_UTR_variant, intergenic region, downstream gene variant, synonymous variant, and splice region variant. For a few patient IDs that were found in more than one dataset, their mutation profiles were merged (if different).

Overall survival data was collected from the published datasets. A few patients with discrepant data (for instance, ICGC_MB193 was 2.3 years old according to Northcott dataset, but 70 years old according to PBCA-DE dataset) were excluded. For 18 samples with different follow-up, we accepted the newest survival time values from Northcott dataset.

Data from all the datasets were combined into one cohort dubbed MB(union), so that 541 patients were covered with both clinical and exome sequencing data.

## TCGA cohorts
The TCGA data were obtained via https://portal.gdc.cancer.gov/. Clinical profiles were used according to the most recent update (*Liu et al., 2019*).

## Network enrichment analysis
Network enrichment between two gene sets of interest $S_a$ and $S_b$ is estimated by comparing the actual number of network edges $\hat{\varepsilon}_{S_a \leftrightarrow S_b}$ that connect nodes of $S_a$ with nodes of $S_b$ in the real, biological network $G_B=(E,V)$ with a number expected by chance $\hat{\varepsilon}_{S_a \leftrightarrow S_b}$ in a random network $G_R=(E,V)$ where particular node degrees $k$ of genes $\forall g_i \in S_a$; $\forall g_j \in S_b$; $g_i \neq g_j$ equal to those of the actual network (which implicitly assumes that the whole degree sequences of $G_B$ and $G_R$ are identical, too). In an earlier work (*Alexeyenko et al., 2012*), series of randomized instances of $G_R$ were created using an algorithm of explicit edge permutation (*Maslov and Sneppen, 2002*) and used for estimating expected variance of $\varepsilon$. Later, it was demonstrated (*Jeggari and Alexeyenko, 2017*) that $\hat{\epsilon_{i \leftrightarrow GS}}$ can be calculated analytically in a fast and unbiased manner:

$$\hat{\varepsilon}_{S_a \leftrightarrow S_b} = \left( \sum_{i=1}^{|S_a|} \mathrm{k}_i * \sum_{j=1}^{|S_b|} \mathrm{k}_j \right) /2|\mathrm{E}|;$$

Then the difference between the actual and expected edge counts

$$\Delta \varepsilon = \varepsilon_{S_a \leftrightarrow S_b} - \hat{\varepsilon}_{S_a \leftrightarrow S_b};$$

is used to estimate significance of the relation $S_a \leftrightarrow S_b$ with a $\chi 2$ statistic:

$$\chi^2 = \frac{\Delta\varepsilon_i^2}{\hat{\varepsilon}_{S_a \leftrightarrow S_b}} + \frac{\Delta\varepsilon^2}{|E| - \hat{\varepsilon}_{S_a \leftrightarrow S_b}},$$

The $\chi2$ does not follow Gaussian distribution, but it can be conveniently converted to Z-scores and then used safely for downstream calculations in for example linear models.

In the simplest NEA case one of the sets is a single gene $i$:

$$\hat{\varepsilon}_{i \leftrightarrow S} = (K_i * \sum_g k_g)/2|E|; \ \ \Delta\varepsilon_i = \varepsilon_{i \leftrightarrow S} - \hat{\varepsilon}_{i \leftrightarrow S};$$

$$\chi^2 = \frac{\Delta\varepsilon_i^2}{\hat{\varepsilon}_{i \leftrightarrow S}} + \frac{\Delta\varepsilon_i^2}{|E| - \hat{\varepsilon}_{i \leftrightarrow S}},$$

which simplified calculation and – within this work – enabled estimation of network enrichment for a mutated gene against the (rest of) mutations in the same cancer genome, called mutated gene set (MGS) in MutSet method or functional gene set (FGS, or simply pathways) in case of PathReg.

## Network

For NEA we merged network of top 1 million edges, ranked by confidence (i.e. Final Bayesian Score) from FunCoup 3 (*Schmitt et al., 2014*) and all edges of Pathway Commons 9 (*Cerami et al., 2010*; *Rodchenkov et al., 2020*). We made sure that all genes reported as altered in at least one of the ten cancer cohorts had up-to-date gene symbols in network. That resulted in a network of 19,035 nodes (unique gene symbols) connected with 1,731,648 unique edges.

## Mutation gene sets and driver gene sets

We defined mutated gene sets (MGS) as lists of all genes of a given tumour sample reported with somatic mutations (SM) in the MAF files. MGSs were used as whole sets in driver evaluation of MutSet channel. MGSs **did not include** copy number altered (CNA) genes.

The analysis included all mutations reported in the TCGA MAF files, regardless of predicted functional impact. Indeed, although synonymous and intronic mutations were often disregarded in cancer research, their involvement in carcinogenesis seems likely and has been recently demonstrated (*Sharma et al., 2019*). In our datasets, frequency of silent mutations was somewhat lower than of non-synonymous ones, but still significant so that many frequently mutated genes showed elevated rates in both categories (*Supplementary file 7*). Therefore, each altered gene $i$ from a given sample, either SM or CNA, was evaluated against the MGS and received a MutSet q-value. Significantly altered genes with were included in final driver gene sets $DGS_{0.05}$ and $DGS_{0.01}$ under conditions $q(MutSet\&PathReg)<0.05$ and $q(MutSet\&PathReg)<0.01$, respectively.

## Functional gene sets

For the PathReg predictor, we used 318 KEGG pathways from version as of 16 August 2018. Considering the importance of SHH and WNT pathways in e.g. medulloblastoma, alongside with respective KEGG pathways we included these two also in Biocarta (*Nishimura, 2001*) versions (the versions were very different in size and length). We updated gene symbols in the sets in the same way as described for the mutations above.

For the survival analysis, the FGS collection consisted of 1,659 entries from BioCarta (*Nishimura, 2001*), KEGG (*Kanehisa et al., 2002*), Reactome (*Croft et al., 2014*), WikiPathways (*Pico et al., 2008*), MetaCyc (*Caspi et al., 2014*), and MSigDB hallmarks (*Liberzon et al., 2015*).

## Altered gene sets (transcriptomics)

The gene expression data was used from the available cohort data sets:

- Affymetrix for MB (*Robinson et al., 2012*);
- Agilent for OV and GBM;
- IlluminaHiSeq_RNASeqV2 for the rest of TCGA cohorts.

The AGS were compiled as sample-specific lists of top $N$ genes ($N=[50,100,200]$) with normalized mRNA expression most different from the respective cohort mean using function samples2ags(…, method = "topnorm") from R package NEArender (*Jeggari and Alexeyenko, 2017*).

## NEAdriver: algorithm

The driver discovery algorithm combined results from two NEA-based channels, MutSet and PathReg.

## MutSet channel

The MutSet values quantified network enrichment between each gene $m$ having a somatic point mutation in genome $j$ and the set MGS of all other point mutations in the same genome ($m \in MGS_j$). They were calculated as NEA scores $Z_{m \leftrightarrow MGS_j}$ so that within a cohort the same gene might receive multiple, sample-specific MutSet values. Respective p-values were obtained from the normally distributed $Z_{m \leftrightarrow MGS_j}$ values with a trivial R function pnorm (**Jeggari and Alexeyenko, 2017**).

## PathReg channel

As the independent variable for training the PathReg predictor, we employed vectors anchor.summary. Specific NEA scores were calculated for every gene $i$ present in the network (N=19035) versus every MGS in the given cancer cohort $c$. The anchor.summary values $\mu_{ic}$ were then obtained by summing up over all $N_c$ available samples, regardless of mutation status of $i$ in genome $j$:

$$\mu_{ic} = \sqrt{\log \frac{\sum_{j=1}^{j \leq N_c} Z_{i \leftrightarrow MGS_j}}{N_c}};$$

Since the score $Z_{i \leftrightarrow MGS_j}$ is derived from the network patterns of mutated genes across the cohort and does not depend on the mutation profile of $i$ itself, the $\mu_{ic}$ value would reflect a general propensity of $i$ to interact with constellations of putative cancer genes. We note that the algorithm is not given any information on previously identified cancer driver genes and works in the assumption that passenger mutations would not produce relevant signal. The transformations via $\chi 2 \rightarrow Z$, log, and square root were imposed in order to render distributions closer to Gaussian.

The $\mu_{ic}$ profiles were rather scarce due to rare occurrence in MGS of true drivers that would interact with a given gene $i$. We thus further improved the gene specific values via modeling $\mu_{ic}$ with pathway NEA scores $Z_{i \leftrightarrow FGS}$. These were calculated for 320 FGS versus each of the $N$ network genes and then used as a matrix of dependent variables $\Phi$ in PathReg training.

Then sparse regression models were created using function cv.glmnet from R package glmnet (**Friedman et al., 2010**).The chosen package implements elastic net models for solving the problem:

$$\min_{\beta_0,\beta} \frac{1}{N} \sum_{i=1}^{N} w_i l \left( y_i, \beta_0 + \beta^T x_i \right) + \lambda \left[ \frac{(1-\alpha) \|\beta\|_{l2}^2}{2} + \alpha \|\beta\|_{l1} \right],$$

where α is a mixing parameter for balance between lasso and ridge regression (whereby $\alpha$=0 and $\alpha$=1 would lead to plain ridge and lasso regressions, respectively). In our case ($\alpha$=1), glmnet solved just the lasso problem:

$$\min_{\beta_0,\beta} R_\lambda \left( \beta_0, \beta \right) = \min_{\beta_0,\beta} \left( \frac{1}{N} \sum_{i=1}^{N} \left( \mu_{ic} - \beta_0 - \beta^T \Phi \right) + \lambda \|\beta\|_{l1} \right)$$

Parameter $\lambda$ determines complexity of the multivariate regression model, i.e. what subset of initially submitted variables of $\Phi$ should receive non-zero coefficients. Under 3-fold cross-validation, function cv.glmnet tested a series of $\lambda$ values while controlling the cross-validation mean squared error (CVM). The cohort-specific optimum $\lambda_c$ was found as a trade-off between model precision and complexity using Bayesian information criterion (BIC), which was deemed preferable (**Giraud, 2015**) over Akaike information criterion in the context of favourable dimensionality ($n_m \gg p; p = 320$). The optimal $\lambda_c$ was set at the number of FGS variables with non-zero coefficients $k$ as the smallest possible within 2 standard errors of BIC from the lowest BIC value:

$$k: \quad \lambda = \arg \min_\lambda \left[ BIC < \left( \inf \left( BIC \right) + \frac{2\sigma_{BIC}}{\sqrt{n}} \right) \right];$$

$$m_{ic} = \beta_0 + \sum_{j=1}^{k} \beta_j f_{ij}; \quad \forall f \in \Phi; |\beta \ni \beta \neq 0| = k;$$

After this training and model selection step, the retained test subsets were used to check how the original values $\mu_{ic}$ correlate with the predicted values $m_{ic}$ (**Supplementary file 1**).

The distribution of $m_{ic}$ values was non-parametric but close to Gaussian. Therefore, respective p-values were modelled via a normal distribution where mean and standard deviation were estimated as median and 84.2th percentile of the empirical distribution, respectively:

$$\bar{m} = \widetilde{m};\ \sigma = P_{84.2}\left(m\right)$$

(in the Gaussian distribution 84.2% of values are within $\bar{m} \pm \sigma$).

## Integration of channels

The p- values from both MutSet and PathReg were adjusted to with Benjamini-Hochberg method (**Benjamini and Hochberg, 1995**). These *q*-values were equivalent to false discovery rate which conveys the probability of a given driver prediction to be false. These values were integrated into the final value as a product *q(MutSet&PathReg)=q*$_{MutSet}$*q*$_{PathReg}$, which presented the probability that neither channel have produced true predictions. Therefore, 1 – *q(MutSet&PathReg)* was the probability of either channel to be true and we convened to trust a driver prediction if *q(MutSet&PathReg)<c (c=*[0.01, 0.05]).

## Gold standard and alternative driver sets

### Literature-based sets

As gold standard for **MB**, we compiled a list of unique 516 gene symbols (Hg19 human genome version), of which 12 were found in OMIM database, 140 in Disease Ontology, and 399 in MB-related publications found in PubMed:

- Cavalli, F. M. G. et al. Intertumoral Heterogeneity within Medulloblastoma Subgroups. *Cancer Cell* **31**, 737–754.e6 (2017).
- Ellison, D. W. et al. Medulloblastoma: clinicopathological correlates of SHH, WNT, and non-SHH/WNT molecular subgroups. *Acta Neuropathol. (Berl.)* **121**, 381–396 (2011).
- Gajjar, A. et al. Pediatric Brain Tumors: Innovative Genomic Information Is Transforming the Diagnostic and Clinical Landscape. *J. Clin. Oncol.* **33**, 2986–2998 (2015).
- Hovestadt, V. et al. Decoding the regulatory landscape of medulloblastoma using DNA methylation sequencing. *Nature* **510**, 537–541 (2014).
- Jones, D. T. W. et al. Dissecting the genomic complexity underlying medulloblastoma. *Nature* **488**, 100–105 (2012).
- Northcott, P. A. et al. Medulloblastomics: the end of the beginning. *Nat. Rev. Cancer* **12**, 818–834 (2012).
- Northcott, P. A. et al. Subgroup-specific structural variation across 1,000 medulloblastoma genomes. *Nature* **488**, 49–56 (2012).
- Northcott, P. A., Dubuc, A. M., Pfister, S. & Taylor, M. D. Molecular subgroups of medulloblastoma. *Expert Rev. Neurother.* **12**, 871–884 (2012).
- Parsons, D. W. et al. The genetic landscape of the childhood cancer medulloblastoma. *Science* **331**, 435–439 (2011).
- Pugh, T. J. et al. Medulloblastoma exome sequencing uncovers subtype-specific somatic mutations. *Nature* **488**, 106–110 (2012).
- Robinson, G. et al. Novel mutations target distinct subgroups of medulloblastoma. *Nature* **488**, 43–48 (2012).
- Taylor, M. D. et al. Molecular subgroups of medulloblastoma: the current consensus. *Acta Neuropathol. (Berl.)* **123**, 465–472 (2012).

We included all altered genes mentioned in these publications. Mechanisms of alteration included changed methylation, gene copy number, and point mutations.

For the TCGA cohorts, we employed the dedicated KEGG pathways:

- BRCA <- hsa05224:Breast_cancer;
- GBM <- hsa05214:Glioma;
- LUAD <- hsa05223:Non-small_cell_lung_cancer;
- LUSC <- hsa05223:Non-small_cell_lung_cancer;
- SKCM <- hsa05218:Melanoma;
- PRAD <- hsa05215:Prostate_cancer;

- PAAD <- hsa05212:Pancreatic_cancer;
- COAD <- hsa05210:Colorectal_cancer;
- BLCA <- hsa05219:Bladder_cancer;
- OV <- hsa05213:Endometrial_cancer, since origins of these two are intertwined (*Merritt and Cramer, 2010*).

## Gene sets from computational analyses

Bailey2018.PANCAN, N=200 (*Bailey et al., 2018*) PanCancer and PanSoftware analysis over 9,423 tumor exomes from 33 of The Cancer Genome Atlas projects using 26 computational tools.

HCD, N=291 (*Tamborero et al., 2013*) discovered drivers in 12 TCGA cohorts (of which six overlapped with our analysis) with a combination of four algorithms that prioritized mutated genes based on mutation rate, functional impact, positional 'hotspots', and specific enrichment in phosphorylation sites.

IntOGen, N=37…73 (*Martínez-Jiménez et al., 2020*) cancer-type-specific lists of cancer genes (per type) predicted by Integrative OncoGenomics (IntOGen) pipeline.

Martincorena-2017, N=369 (*Martincorena et al., 2017*) a list of known cancer genes used in the molecular evolution (positive selection) analysis and applied 7664 tumors across 29 cancer types.

MuSiC, N=127 (*Kandoth et al., 2013*) identified drivers based on relative point mutation frequency, assisted with expression analysis and database annotations in 12 TCGA cancers (of which six overlapped with our analysis). We used the 'PanCancer' list.

MutSig, N=260 (*Lawrence et al., 2015*) performed a comprehensive point mutation frequency analysis using exome sequencing data from 21 cancer cohorts (of which nine overlapped with our study), while accounting for mutation burden, clustering, and functional impact.

NetSig5000, N=62 (*Horn et al., 2018*) gathered evidence for potential driverness of each gene via functional coupling to frequently mutated genes in the global network. The genes' own mutation frequencies were incorporated into NetSig scores at a separate step.

## Database gene sets

Cancer Gene Census N=12…64 (*Futreal et al., 2004*) cancer type specific lists were downloaded on 7th of February, 2019.

KEGG#05200:Pathways_in_cancer, N=395 (*Kanehisa et al., 2002*) was a 'pan-cancer' version including a curated selection of organ-specific cancer pathway gene lists.

Five 'general' cancer pathways, N=457 (*Kanehisa et al., 2002*) was created as a union of the following cancer related KEGG pathways:

- hsa05202:Transcriptional_misregulation_in_cancer;
- hsa05203:Viral_carcinogenesis;
- hsa05204:Chemical_carcinogenesis;
- hsa05205:Proteoglycans_in_cancer;
- hsa05206:MicroRNAs_in_cancer.

FoundationOne, N=330 (https://www.foundationmedicine.com/resources) is the targeted sequencing panel used for cancer diagnostics, created as a merge of 'general' and 'rearrangements' sections.

## Gene sets from individualized analyses

OncoIMPACT (N=162…695 per cohort) (*Bertrand et al., 2015*) used an expression-driven approach to verify driver roles of point mutations in five TCGA cohorts (of which four overlapped with our analysis). The paper reported cohort-specific driver ranks rather than individual, sample-level estimates. The genes that received a rank were used in comparisons with NEAdriver results.

SCS (*Guo et al., 2018*) reported driver role evaluation in individual samples. They reported only top 50 genes after global, cohort-specific ranking. In parallel, the tables contained top 50 genes from OncoIMPACT (*Bertrand et al., 2015*), DriverNet (*Bashashati et al., 2012*), DawnRank (*Hou and Ma, 2014*), and HotNet2 (*Reyna et al., 2018*; *Leiserson et al., 2015*) methods, which we also imported and used in our comparison.

## Sets of artefactual driver genes

Lawrence2013.FPs, N=19 (*Lawrence et al., 2013*) list of genes frequently presented in literature as false positive cancer drivers.

Martincorena2017.artifacts, N=49 (*Martincorena et al., 2017*) list of genes which are usually heavily-affected by sequencing artifacts.

IntOGen.KnownArtifacts, N=19 (*Martínez-Jiménez et al., 2020*) list of genes labelled as "Known artifact" in their resulting table.

## Normalization of mutation frequencies

The number of samples where each given gene was mutated was normalized by dividing it with its CDS length. When necessary, the sample-specific total mutation burden values were accounted for as total number of point mutations reported in all genes per sample.

## Code availability

Documented code is available under the modified BSD license on GitHub: https://github.com/avev-iort/NEArender-2.x, (copy archived at swh:1:rev:5829beb819c689790359f199547362a31d1a1d54; *Petrov, 2022*).

## Acknowledgements

The authors are grateful to Vetenskapsrådet for provided funding. The analysis used data generated by the TCGA Research Network: https://www.cancer.gov/tcga.

## Additional information

### Funding

| Funder | Grant reference number | Author |
|---|---|---|
| Vetenskapsrådet | 2016-04940 | Iurii Petrov<br>Andrey Alexeyenko |

The funders had no role in study design, data collection and interpretation, or the decision to submit the work for publication.

### Author contributions

Iurii Petrov, Data curation, Formal analysis, Investigation, Resources, Software, Validation, Writing - review and editing; Andrey Alexeyenko, Conceptualization, Funding acquisition, Investigation, Methodology, Software, Supervision, Visualization, Writing – original draft

### Author ORCIDs

Andrey Alexeyenko (iD)http://orcid.org/0000-0001-8812-6481

### Decision letter and Author response

Decision letter https://doi.org/10.7554/eLife.74010.sa1
Author response https://doi.org/10.7554/eLife.74010.sa2

## Additional files

### Supplementary files

• Supplementary file 1. Performance of PathReg multiple regression models. Observed vs. predicted values of anchor.summary values represents performance of created models on continuous scale. In absence of a strict cut-off, performance was measured as a correlation between anchor.summary observed for each gene in the given cohort versus the a value predicted by the multiple regression model. In heatmaps, values next to gene names indicate number of samples with mutations in the given gene.

• Supplementary file 2. Size vs node degree of cancer genes included in NEAdriver driver

predictions, compared against the union of all alternative sets. X coordinates for the alternative sets represent their sizes. For NEAdriver (which in total, across all cohort samples typically predicted hundreds genes, most being very rare) the sets represent samples n=[50, 100, 200, 400] genes most frequently predicted in each cohort.

• Supplementary file 3. Comparison of NEAdriver results with OncoIMPACT (4 cohorts).

• Supplementary file 4. Comparison of NEAdriver results with other network-based methods (4 cohorts, 50 top genes from each method).

• Supplementary file 5. Dependence of NEAdriver q-values from covariates. The five pages present relations between MutSet&PathReg q (Y axis) and no. of mutations per cohort, gene length, and normalized mutation frequency, replication rate, and gene expression rates, respectively (X axis). Top left legend: Spearman rank R and Kendall tau represent overall correlations between X and Y coordinates regardless of other factors. Bottom left legend: terms' significance in 4-way linear models. Colored points: genes suggested as potential artifacts in literature; those receiving q<0.05 are text-labeled.

• Supplementary file 6. Kaplan-Meier plots for 10 cohorts: different survival metrics, dichotomized by NEA scores for either DGS or GE AGS.

• Supplementary file 7. No of mutations versus normalized mutation frequency: relative frequencies of silent and non-silent mutations per gene.

• Supplementary file 8. Summary tables over each of the ten cohorts. PathReg, MutSet, and combined values per sample, per gene, in each cohort. MutSet q-values are accompanied with no. of network links observed between the given gene and point mutations in the sample, as well as respective NEA Z and NEA p-value. All the MutSet values are sample-specific. PathReg q-values are accompanied with respective anchor.summary, PathReg score, and PathReg p-values. All the PathReg values are cohort-specific and not sample-specific. NEAdriver q-value is product of MutSet q-value and PathReg q-value. The last 3 columns indicate genes listed as possible artifacts in literature.

• Transparent reporting form

### Data availability

All data generated or analysed during this study are included in the manuscript and supporting files.

The following previously published dataset was used:

| Author(s) | Year | Dataset title | Dataset URL | Database and Identifier |
|---|---|---|---|---|
| TCGA consortium | 2008 | The Cancer Genome Atlas | https://www.cancer.gov/tcga | TCGA, Genome-Atlas |

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
