## [Editor Report]

In this work, Petrov and Alexeyenko present a novel network-based method to infer cancer driver genes that is not based on frequency of mutations, NEAdriver, and evaluate its performance across a large dataset. This manuscript addresses a topic of high interest in the cancer genomics community and is a welcome addition to the literature.

---

## [Decision Letter]

**Decision letter after peer review:**

Thank you for submitting your article "Individualized discovery of rare cancer drivers in global network context" for consideration by *eLife*. Your article has been reviewed by 3 peer reviewers, one of whom is a member of our Board of Reviewing Editors, and the evaluation has been overseen by Aleksandra Walczak as the Senior Editor. The following individual involved in review of your submission has agreed to reveal their identity: Abel Gonzalez-Perez (Reviewer #2).

Essential revisions:

1. The three reviewers agreed that the authors should provide a reasonable calculation of the rate of false positives that their method produces, as at the moment is hard to evaluate what this number may be. The authors do not seem to correct for the background mutation rate of genes in the calculation of MutSet. This may result in identifying false positive driver genes with abnormally high number of mutations across cohorts due to known covariates of the mutation rate, such as the replication time or the level of transcription. Correctly estimating the background mutation rate of genes across tumors is a key tenet of methods that search for signals of positive selection in genes. For further explanation on this subject see PMID 23770567 and PMID 29056346. This may explain why known highly mutated non driver genes like TTN, RYR1 and others appear as significant recurrently across different cohorts. This issue is key for any method that uses mutation data to identify driver genes and must be addressed by the authors. A reviewer also adds adds that one way to calculate the rate of false positives would be to compile a list of known artifactual genes identified by methods aiming at detecting signals of positive selection in their mutational pattern across tumors. Computing the overlap of the output of their method with this list across cohorts could provide such estimate. An example of such a list is also provided in the supplementary material of this paper: PMID 32778778.

2. Reviewers also noted that the paper would benefit from a clearer presentation of the algorithm. Specifically:

a) Have the authors performed any technical filtering of the mutations? If not, how can they ensure that their results are not affected by potential sequencing artifacts or germline contamination?

b) The authors state that "The procedure evaluated likelihood of each genomic alteration (either PM or CNA) being a driver in each tumor genome." However, apparently results are only provided at the level of gene-cohort. Can this be addressed, please?

c) How are different sample-specific MutSet values converted into a single cohort-wise value per gene?

d) Is the only difference between the MutSet channel and the summary computed in PathReg channel the fact that in the former the NEA is computed only for mutated genes, while in the latter it is computed for all genes in the network?

e) When explaining the PathReg, the authors state that "the μic value would reflect a general propensity of i to interact with constellations of putative cancer genes" and that "The μic profiles were rather scarce due to rare occurrence in MGS of true drivers that would interact with a given gene i". However, in the μic measures the NEA score of each gene with the mutated genes across samples, not with previously identified driver genes. Can the authors clarify this please?

f) How are p-values for the MutSet algorithm computed?

g) How were the cutoff qvalues of 0.05 and 0.01 selected? How do they compare in terms of number of genes that pass each?

h) Could the authors clarify the rationale under the selection of nine TCGA cohorts (among 34) to apply their algorithm?

i) It is not clear to some reviewers why the authors described their algorithm as "individualized". It may give the impression that it is aimed at identifying alterations driving a patient's tumor, rather than genes that are cancer drivers across tumors. Could this terminology be clarified please?

3. Please articulate more clearly what the conclusions of the study are. Results should provide a big picture summary of the method that is understandable to non-experts, while technical details might be better presented in the methods. Specifically, it would be good to answer the following questions:

a) How many driver genes are identified in each cohort? This information does not seem easy to find. A long supplementary Table contains q-values for all genes (>19,000) across all cohorts, but this is impractical to navigate. It would be easier to have a table containing only significant genes, with the numbers reported in the text and presented in Figures.

b) It was brought up that KEGG Pathways in Cancer may not be a good standard to evaluate the performance of an algorithm aimed at identifying driver genes from their mutations across tumors. Reasons are: First, they contain manually selected genes that fit functional interaction in pathways, which may leave out some relevant genes. Second, because they contain genes connected through functional interactions into pathways, some of them may not be drivers at all, but only connected to drivers. Since the NEAdriver method exploits network connections to identify significant genes it may not be surprising that its output shows the best overlap with these curated pathways. It may be better to do a more comprehensive comparison with gold standard lists of cancer driver genes, such as the Cancer Gene Census (this would be solved by doing the false positive rate calculation described in point 1).

c) What is the gold standard set used to compute the ROC curves shown in Figure 2? From the text it would seem that genes in KEGG pathways are used as true positives, but these are also evaluated, so the evaluation dataset must be external. Can the authors please clarify this?

d) There's a plethora of published methods that identify signals of positive selection in the mutational pattern of genes across tumors. For articles that summarize groups of such methods, see PMID 29625053 or PMID 32778778. This paper should include a thorough benchmark and comparison of the NEAdriver method to these state of the art algorithms.

e) Please specify which version of the genomics data was was used (the gcs.cancer.gov portal hosts multiple versions from the TCGA data -Hg19 and Hg38, multiple file formats). Linked to this – why did the authors not use a more recently processed version of the TCGA data than the one on GDC portal (e.g. there are multiple pan-cancer follow up publications)? And, could it be specified what type of data was downloaded from TCGA (VCFs, MAFs, Or BAMs)? This is not specified in the methods and without this information it is hard to assess what the authors have done to ensure that their variant calling is accurate. For example, a reviewer mentioned that there are a number of similar genes in the heatmaps (ITGB6, ITGA8, ITGAL etc). These may act in the same pathways but are also paralogues – how can the authors be sure that mutations detected in these are real and not an issue of mismapping?

f) Finally, more generally, is there a new class of driver genes that can be identified based on the authors' approach that could not be understood based on previous studies, or alternatively, could this method/strategies be expanded to predict other phenotypes than driver genes? Was a new cancer gene or driver mutation discovered that could not be explained previously and that can now be viewed in a new perspective based on the authors' work? Highlighting a few of these examples would make the impact of their paper much stronger.

4. Please add a code availability section (e.g. GitHub repository).

---

## [Author Response]

Essential revisions:1. The three reviewers agreed that the authors should provide a reasonable calculation of the rate of false positives that their method produces, as at the moment is hard to evaluate what this number may be. The authors do not seem to correct for the background mutation rate of genes in the calculation of MutSet. This may result in identifying false positive driver genes with abnormally high number of mutations across cohorts due to known covariates of the mutation rate, such as the replication time or the level of transcription. Correctly estimating the background mutation rate of genes across tumors is a key tenet of methods that search for signals of positive selection in genes. For further explanation on this subject see PMID 23770567 and PMID 29056346. This may explain why known highly mutated non driver genes like TTN, RYR1 and others appear as significant recurrently across different cohorts. This issue is key for any method that uses mutation data to identify driver genes and must be addressed by the authors. A reviewer also adds adds that one way to calculate the rate of false positives would be to compile a list of known artifactual genes identified by methods aiming at detecting signals of positive selection in their mutational pattern across tumors. Computing the overlap of the output of their method with this list across cohorts could provide such estimate. An example of such a list is also provided in the supplementary material of this paper: PMID 32778778.

At the initial stage of this project, we spent much time investigating associations between frequency and other input or background characteristics and the NEAdriver. Somewhat surprisingly but consistently with nature of the algorithm, we did not find any strong relations. Unfortunately, we took this as self-evident and never reported it in the manuscript. Now we did our best to prove that it indeed did not significantly affect the reported results. In section “Rates of driver discovery rate versus mutation frequency and possible confounders” we now presented an in-depth analysis of NEAdriver results in relation to the factors pointed by reviewers.

From the papers which you recommended, we obtained lists of known or suggested artefactual genes: Martincorena et al., (2017), Lawrence et al., (2014), and Martínez-Jiménez (2020), as well as the whole category of olfactory receptors mentioned as implausible drivers by e.g. Lawrence et al., (2014). These genes were color- and text-marked in the new analysis and included as a co-factor in the linear model analyses (see also below). Further, similarly to Figure 3 by Lawrence et al., (2013) we evaluated influence of known confounders (gene length, replication rate, expression level, and total mutation burden per sample) on the NEAdriver predictions in 4-way linear models. While Lawrence and co-authors demonstrated strong correlations of mutation rate versus replication or expression, respective correlations of NEAdriver score were not strong at all. NEAdriver predictions stronger associated with CDS length – but the same tendency was noted in all the analyzed gene sets (see new Figure Supplement to Figure 4). Also, genes of all the computational sets more frequently contained point mutations and had higher mutation frequency per b.p. length, both absolute and normalized by genome-specific mutation load. The same was also true for the driver set by Lawrence et al., (2014), which supposedly would be least affected by these factors. Otherwise, probability of being predicted by NEAdriver after adjustment for gene length and TMB proved to be weak.

New Suppl. File 5 displays artifactual genes which surpassed significance threshold q<0.05, so that we see 1…13 genes per cohort. Interestingly, our previous estimates of false discovery rates with 2 alternative methods ranged between 20 and 80% – while only 0.5…3% of the NEAdriver predictions were found in the artifact gene lists (upper right legends in Suppl. File 5). For comparison, the same genes made up 1…9% of the alternative computational sets.

Lawrence et al., (2013) pointed that several “false positives” had cropped up and rooted in cancer genome studies. In Discussion, we considered in detail one such gene, titin and provided alternative explanations of how, despite the apparently high length, it might still be involved in carcinogenesis. Therefore, we decided to not exclude any artifactual genes suggested in the mentioned publications but instead labeled them in last columns of the new final result tables (Supplementary File 8). Also, Figure 2B is now made upon exclusion of such genes from all the sets (which did not lead to any significant changes in the tree topology).

2. Reviewers also noted that the paper would benefit from a clearer presentation of the algorithm. Specifically:a) Have the authors performed any technical filtering of the mutations? If not, how can they ensure that their results are not affected by potential sequencing artifacts or germline contamination?

Thank you for this question. The mutation sets per sample were used from MAF files as delivered by TCGA (level 3 data) and from MAF files or curated lists for MB related projects. This setting corresponds to the suggested role of NEAdriver algorithm, which is evaluation of any reported mutations and filtering out passenger ones based on network connectivity, similarly to real-life clinical evaluation of in a molecular tumor board.

b) The authors state that "The procedure evaluated likelihood of each genomic alteration (either PM or CNA) being a driver in each tumor genome." However, apparently results are only provided at the level of gene-cohort. Can this be addressed, please?

Indeed, for the sake of reducing file size we decided to submit the resulting table in a “compact” format, i.e. per gene&cohort rather than per sample&gene&cohort. Now (see also the next question) we have instead generated and submitted separate, per cohort Excel files which contain ~1 million rows in total (Suppl. File 8).

c) How are different sample-specific MutSet values converted into a single cohort-wise value per gene?

Thank you for this question. For the initial submission we summarized the MutSet values with Fisher’s formula for combining p-values, which was described only as a comment in the Excel table header – likely not noticeable. Now we realized that this way of presentation was extremely unfortunate. Beyond raising the current question, the Fisher’s approach was misleading and resulted in an incorrect presentation of final NEAdriver predictions, since these values were then multiplied with PathReg p. We also noticed that it emphasized the artefactual genes much stronger than they deserved by their NEAdriver scores. This calculation was only applied in this table and is now fully abandoned. Instead, we presented the genuine NEAdriver results in full tables (see answer to the previous question).

d) Is the only difference between the MutSet channel and the summary computed in PathReg channel the fact that in the former the NEA is computed only for mutated genes, while in the latter it is computed for all genes in the network?

No, the major difference is that the MutSet scores are limited to own sample’s mutations, whereas the PathReg score for the same gene would be based on a sum of such scores against ALL samples, including those where it was wild type. So PathReg is computed “across samples, via external genes and pathways” rather than “within a specific mutated gene set”. As a result, MutSet and PathReg are quite uncorrelated (see also the next reply). We emphasized this difference in the revised version.

e) When explaining the PathReg, the authors state that "the μic value would reflect a general propensity of i to interact with constellations of putative cancer genes" and that "The μic profiles were rather scarce due to rare occurrence in MGS of true drivers that would interact with a given gene i". However, in the μic measures the NEA score of each gene with the mutated genes across samples, not with previously identified driver genes. Can the authors clarify this please?

Thank you for pointing to this part which indeed required clarification. The idea was that PathReg gathers scarce information from *implicit* drivers (which would be very few per sample) without input from previous knowledge. All mutations have *a priori* equal weights, but the signal from passengers should be negligible and is filtered out. We suppose that this “explicit vs. implicit” difference was unexplained and caused the misunderstanding. We elaborated on this in the text. We should also mention that we tested an extra channel for NEAdriver against most frequent mutated genes (similarly to NetSig algorithm) but it did not produce any extra benefit for NEAdriver performance and was not used.

f) How are p-values for the MutSet algorithm computed?

Thank you for noticing this, we missed to explain it. Briefly, the normally distributed (under null) NEA z-scores were converted to p-values with a standard R function (pnorm). We added this to Methods now.

g) How were the cutoff qvalues of 0.05 and 0.01 selected? How do they compare in terms of number of genes that pass each?

The cutoff levels affected discovery rates rather weakly. In particular, the agreement with gold standard sets was satisfactory already at q<0.2 and did not drop steeply after that (Figure 3C,D,E,F). Thus the two q-value cutoffs were chosen quite arbitrarily, just to provide a certain alternative. The new Table 1 allows comparing the cutoff effects on the number of genes. Under q<0.05 the fractions of drivers were 14…67% larger than under q<0.01. We also mentioned it in the text (page 5).

h) Could the authors clarify the rationale under the selection of nine TCGA cohorts (among 34) to apply their algorithm?

This work was not a part of TCGA or a PanCancer project. Our major purpose of having multiple cohorts was to investigate the method’s efficiency over a range of typical mutation frequencies per sample. By the moment of project start, we selected TCGA cohorts with highest number of samples with exome-seq and other potentially relevant data. Medulloblastoma was introduced for having least somatic mutations per sample.

i) It is not clear to some reviewers why the authors described their algorithm as "individualized". It may give the impression that it is aimed at identifying alterations driving a patient's tumor, rather than genes that are cancer drivers across tumors. Could this terminology be clarified please?

By using this word we referred to the classification of driver discovery methods into those collecting cohort-wise statistics and those looking at individual patient/sample features, NEAdriver belongs to the latter category. So, the impression was actually correct, but it is just “individualized” rather than fully “individual” because PathReg, while also using individual samples, still produces cohort-wise scores. We now elaborated on this in Abstract.

3. Please articulate more clearly what the conclusions of the study are. Results should provide a big picture summary of the method that is understandable to non-experts, while technical details might be better presented in the methods. Specifically, it would be good to answer the following questions:a) How many driver genes are identified in each cohort? This information does not seem easy to find. A long supplementary Table contains q-values for all genes (>19,000) across all cohorts, but this is impractical to navigate. It would be easier to have a table containing only significant genes, with the numbers reported in the text and presented in Figures.

Thank you for this suggestion. We have created a new Table 1 with counts and percentages of unique genes and mutations per cohort predicted with the channels and altogether.

b) It was brought up that KEGG Pathways in Cancer may not be a good standard to evaluate the performance of an algorithm aimed at identifying driver genes from their mutations across tumors. Reasons are: First, they contain manually selected genes that fit functional interaction in pathways, which may leave out some relevant genes. Second, because they contain genes connected through functional interactions into pathways, some of them may not be drivers at all, but only connected to drivers. Since the NEAdriver method exploits network connections to identify significant genes it may not be surprising that its output shows the best overlap with these curated pathways. It may be better to do a more comprehensive comparison with gold standard lists of cancer driver genes, such as the Cancer Gene Census (this would be solved by doing the false positive rate calculation described in point 1).

Thank you for these suggestions. The KEGG pathways were used as a reference which is both human-curated and cancer site-specific (and we did believe that having genes “manually selected” and “only connected to drivers” would be an advantage). But now we have complemented the references with the suggested site-specific Cancer Gene Census, Martincorena et al., (2017) as well as IntOGen genes sets. In some cohorts, using CGC as a standard produced even more optimistic results than KEGG.

c) What is the gold standard set used to compute the ROC curves shown in Figure 2? From the text it would seem that genes in KEGG pathways are used as true positives, but these are also evaluated, so the evaluation dataset must be external. Can the authors please clarify this?

We assume Figure 3 was meant here (Figure 2 is a heatmap). In addition to the single reference set in Figure 3E,F we now employed benchmarks based on Cancer Gene Census, IntOGen, and the list of “369 known cancer genes” (Martincorena et al., 2017). These now appear as novel lines in Figures 3E and F (the dataset descriptions can be found in Methods). Also, the legend was probably misleading. Under “KEGG” here we meant site-specific cancer pathways for the cohorts rather than ALL pathways. When not available from KEGG (as for MB cohort), custom literature-based sets were used. Since there was no way to find an absolutely external, non-overlapping set for this analysis, in Figure 3A,B,C,D we used as many as 12 different gene sets.

d) There's a plethora of published methods that identify signals of positive selection in the mutational pattern of genes across tumors. For articles that summarize groups of such methods, see PMID 29625053 or PMID 32778778. This paper should include a thorough benchmark and comparison of the NEAdriver method to these state of the art algorithms.

We have now included both 29625053 (Bailey et al., 2020) and 32778778 (IntOGen) in the analysis. We also note that many of the methods which MC3 project combined (Bailey et al., 2018) had been analyzed individually in our manuscript.

e) Please specify which version of the genomics data was was used (the gcs.cancer.gov portal hosts multiple versions from the TCGA data -Hg19 and Hg38, multiple file formats). Linked to this – why did the authors not use a more recently processed version of the TCGA data than the one on GDC portal (e.g. there are multiple pan-cancer follow up publications)? And, could it be specified what type of data was downloaded from TCGA (VCFs, MAFs, Or BAMs)? This is not specified in the methods and without this information it is hard to assess what the authors have done to ensure that their variant calling is accurate. For example, a reviewer mentioned that there are a number of similar genes in the heatmaps (ITGB6, ITGA8, ITGAL etc). These may act in the same pathways but are also paralogues – how can the authors be sure that mutations detected in these are real and not an issue of mismapping?

Thank you for this remark. We used MAF files based on Hg19. Regarding the TCGA data, we considered GDC being the most official source, enabling most uniform data processing and used as such by many (all?) PanCancer projects. Although for clinical data, which significantly changed over the years, we did use a PanCancer update by Liu et al., (2019). These details are now provided in the manuscript.

f) Finally, more generally, is there a new class of driver genes that can be identified based on the authors' approach that could not be understood based on previous studies, or alternatively, could this method/strategies be expanded to predict other phenotypes than driver genes? Was a new cancer gene or driver mutation discovered that could not be explained previously and that can now be viewed in a new perspective based on the authors' work? Highlighting a few of these examples would make the impact of their paper much stronger.

Thank you for this suggestion. We now introduced (see last section of Results) a presentation of quite extensive groups of genes that were frequently predicted (in about half of the studied cohorts): integrins, laminins, and collagens. Despite being mutated in nearly each genomic sample, such genes were so far mostly overlooked, probably because of their “non-signaling” roles. We took extreme care of making sure that these are not artifacts and indeed can provide a novel view angle at carcinogenesis.

4. Please add a code availability section (e.g. GitHub repository).

Thank you for bringing this into our attention, code availability section was added.